

# Symmetry-protected gates of Majorana qubits in a high-$T_c$ higher-order topological superconductor platform

**Matthew F. Lapa[1], Meng Cheng[2] and Yuxuan Wang[3⋆]**

**1** Kadanoff Center for Theoretical Physics, University of Chicago, Chicago, IL 60637, USA
**2** Department of Physics, Yale University, New Haven, CT 06520, USA
**3** Department of Physics, University of Florida, Gainesville, FL 32611, USA

⋆ yuxuan.wang@ufl.edu

## Abstract

We propose a platform for braiding Majorana non-Abelian anyons based on a heterostructure between a *d*-wave high-$T_c$ superconductor and a quantum spin-Hall insulator. It has been recently shown that such a setup for a quantum spin-Hall insulator leads to a pair of Majorana zero modes at each corner of the sample, and thus can be regarded as a higher-order topological superconductor. We show that upon applying a Zeeman field in the region, these Majorana modes split in space and can be manipulated for braiding processes by tuning the field and pairing phase. We show that such a setup can achieve full braiding, exchanging, and arbitrary phase gates (including the $\pi/8$ magic gates) of the Majorana zero modes, all of which are robust and protected by symmetries. As many of the ingredients of our proposed platform have been realized in recent experiments, our results provide a new route toward universal topological quantum computation.



# 1   Introduction

In the past two decades, topological quantum computation has attracted great interest in the condensed matter community. They key ingredient of this idea is to encode and manipulate quantum information using non-Abelian anyons, which are inherently non-local degrees of freedom and are thus immune to local error at the hardware level. One of the most promising platform for the physical realization of non-Abelian anyons is topological superconductors that host Majorana Zero Modes (MZM) at boundaries and defects [1–8]. Adiabatic braiding and exchange of the MZMs generate Clifford gates in a topologically protected manner [9, 10], and implementations of such operations have been studied in various platforms [11–15]. However, one drawback of the Majorana platform is that the Clifford gates are not powerful enough to achieve universal quantum computation [16]. It is well known that additional gates (i.e., the magic gate of a $\pi/8$ phase rotation) must be supplemented to achieve universality, which however require non-topological operations. A number of proposals to implement the magic gate in the Majorana platform have been put forward [17–21], most of which rely on precise control over non-universal couplings to essentially realize an arbitrary phase rotation

(a notable example of a robust magic gate using geometric decoupling was proposed in Ref. [20, 21]).

Recently, the concept of topological insulators and superconductors has been generalized to higher-order topological insulators and superconductors [22–55]. Protected by crystalline symmetries [39], higher-order topological superconductors host MZMs at the corners in two spatial dimensions and Majorana modes at the hinges or vertices in three dimensions. With the flourishing ideas on the realization of higher-order topological superconductors, it is natural to search for new possibilities of manipulating Majorana modes using a higher-order topological superconductor. For example, in a recent work [56], the authors proposed a protocol through the manipulation of the Zeeman field and the pairing order parameter, a full braid (corresponding to $\pi/2$ rotations) between a pair of MZMs can be achieved (see also Ref. [57]). In another proposal [58, 59], the authors showed that the exchange of MZMs can be achieved through a multi-step process by tuning three independent Zeeman fields, a protocol similar to that in a T-junction of superconducting nanowires [11].

In this work, we propose a different setup in a higher-order topological superconductor that allows for a much richer set of non-Abelian rotations in the Hilbert space of Majoranas zero modes, including Clifford and symmetry-protected phase gates for MZMs. Our proposed setup is based on several recent works [60, 61] showing higher-order topological superconductors can be achieved in a heterostructure involving a (first-order) topological insulator and unconventional high-$T_c$ superconductors coupled via superconducting proximity effect. In particular, we focus on a heterostructure between a $d$-wave high-$T_c$ superconductor, for example the Bi based cuprate $Bi_2Sr_2CaCu_2O_{8+\delta}$ (BSCCO) that has recently been realized in monolayers [62], and a quantum spin Hall insulator, such as $WTe_2$ [63–65]. For a $d$-wave cuprate superconductor, the pairing symmetry enforces the proximity-induced gap to vanish along the certain directions. When such a pairing gap is induced on the helical edge states of the underlying quantum spin Hall insulator, it creates a Majorana mass domain wall at each corner, thus hosting *two* MZMs. In the context of higher-order topology, the corner Majorana modes are protected by mirror reflection symmetries together with time-reversal symmetry and particle-hole symmetry. The mirror symmetries pin the MZMs at high-symmetry directions, which form a Kramers pair.

For our purposes, however, the model-specific mirror symmetries are unnecessary, and in fact intentionally broken by external control fields, so that the corner MZMs can move along the edge. Instead, we identify two emergent symmetries, an effective time-reversal and a chiral (an anti-unitary charge conjugation) symmetry of the low-energy edge theory, which protect the MZMs even when they are away from the mirror symmetric locations. Using a bosonized edge theory, we determine the localization length and the excitation gap of the MZMs in the presence of interaction effects, which are consistent with the celebrated Kosterlitz-Thouless scaling for infinite systems. Interestingly, even when the spatial profiles of the MZM become large and overlap, their degeneracy remain protected by these symmetries. These additional emergent symmetries also circumvent a no-go theorem [66] that would have allowed local time-reversal-invariant perturbations to spoil the universal non-Abelian Berry phases from braiding a Kramers pair of MZMs.

The key additional ingredient in our platform is an in-plane Zeeman field. To this end, we note that recently, heterostructures involving two dimensional ferromagnets fabricated via molecular-beam epitaxy has already been shown [67] to realize topological superconductivity [68]. In the presence of a Zeeman field, the physical time-reversal symmetry of the quantum spin Hall insulator is broken. However, we show that the emergent *effective* time-reversal and chiral symmetries are still intact, protecting the MZMs. Since they are no longer Kramers partners, the MZMs can split spatially. By tuning the Zeeman field, the position of the Majorana modes can be manipulated. We show that this can be utilized to achieve vari-

ous non-Abelian rotations within the degenerate ground state subspace. First, we show that rotating the in-plane Zeeman field by $2\pi$ is equivalent to a full braid between the two MZMs, which is analogous to previous proposals. Second, as the main result of this work, we demonstrate that by taking the in-plane Zeeman field **B** through a "half-moon" contour in the $B_x$-$B_y$ plane that crosses $B = 0$ (see Fig. 3), one can achieve an exchange process of the two MZMs localized in the same corner, resulting in the hallmark non-Abelian exchange statistics of the Ising anyons. Crucially, we show that the non-Abelian Berry phase of this exchange process is protected by the *physical* time-reversal symmetry broken only by the Zeeman fields, robust against local perturbations. Additionally, we show that dual to this process, one can tune the phase of the complex superconducting order parameter along one edge of the sample to go through the same "half-moon" contour in the complex plane, and achieve the exchange of two MZMs from adjacent corners. The Berry phase during this process is protected by the emergent chiral symmetry. The combination of these two exchange processes realize the Clifford gates in a qubit formed by four MZMs in two adjacent corners. Notably, a finite sample of our setup realizes three qubits, with a set of Clifford gates available on each edge. Third, we show that by going through a "slice of pie" contour (see Fig. 4), the Zeeman field (and analogously the superconducting field) can perform an arbitrary phase gate of the Majorana qubit. This includes the long-sought-after "magic gate" for MZMs, crucial for universal topological quantum computing. Remarkably, the Berry phases in this process are protected by U(1) symmetries (which can be exact or emergent), and hence are robust against random errors as long as the input for the phase angle is sufficiently precise.

Our proposal has several advantages. First and foremost, the high-$T_c$ superconductor platform ensures a higher operating temperature, a larger critical Zeeman field, and better localization of the Majorana modes. As we mentioned, BSCCO and WTe$_2$ are readily available 2d materials for $d$-wave superconductivity and quantum spin Hall effect. In particular, WTe$_2$ has been demonstrated [64, 65] to have a U(1) spin axis needed for our purposes. Second, our protocols of exchanging MZMs consist of simple manipulations of Zeeman or pairing fields, which do not require physically moving around superconducting vortices or tuning multiple parameters in each exchange process. Third, our setup can achieve a universal phase gate protected by symmetries, including the $\pi/8$ magic gate, and thus holds promises for universal topological quantum computation.

While the higher-order topological superconductor platform provides a feasible realization of our proposal, our results are established within the framework of the universal effective field theory description of the topological edge states, which can then be straightforwardly adapted to other systems with the same low-energy description. For instance, we note that the corner MZMs have been shown to exist in similar platforms with iron-based high-$T_c$ superconductors [46, 61]. In general, all of our results can be easily applied to MZMs realized at domain walls between magnetic and superconducting regions on the edge of a quantum spin Hall insulator.

Our analysis can also be directly extended to interacting topological phases with fractional statistics in the bulk. We show that $\mathbb{Z}_{2m}$ parafermion modes [69–73] can be realized using a similar setup with a fractional quantum spin Hall insulator. A key difference from the Majorana case is that, here there are $m-1$ independent dynamical phases that accompanies the non-Abelian Berry phases. Even though the non-Abelian phase is not topologically protected against unitary errors, for small $m$ it may be possible to precisely control the time of operation to tune these dynamical phases to zero. Interestingly, evidence for parafermions have been observed in a similar setup with fractional quantum Hall states in the presence of superconductivity [74].

The remainder of this paper is organized as follows. In Sec. 2 we describe the setup of our proposed platform that hosts pairs of MZMs at its corners. In Sec. 3 we reformulate the

derivation of the corner Majorana modes using a bosonized language, which enables the inclusion of interaction effects and a transparent interpretation of the non-Abelian Berry phases. As the main result of this work, in Sec. 4 we show that such a setup allows symmetry protected Clifford gates and phase gates utilizing the MZMs by tuning an in-plane magnetic field and the phase of the superconducting order parameter. In Sec. 5 we generalize our setup to that with a fractional quantum spin Hall insulator with $m \neq 1$, and show that the Berry phase accumulated using the same protocol corresponds exactly to the exchange statistics of $\mathbb{Z}_{2m}$ parafermions.

We include various details in the Appendices. In Appendix A we present an example of a lattice model for the setup that is a higher-order topological superconductor protected by mirror symmetries and time-reversal symmetry. In Appendices B, C, D, E, and F we present a detailed analysis of the bosonization procedure for our setup and the non-Abelian Berry phases we obtained in a more heuristic manner in the main text. In Appendices G and H we prove the twofold ground state degeneracy at each corner corresponds to the Majorana doublet, from both an operator algebra approach and 't Hooft anomaly perspective.

## 2 Corner MZMs in a high-$T_c$ superconductor platform

Our platform is based on several recent proposals [60,61] of higher-order topological superconductivity realized in a heterostructure formed by a quantum spin Hall insulator (QSH) and a high-$T_c$ $d$-wave superconductor, coupled via superconducting proximity effect. As is well-known, a single-band $d$-wave superconductor hosts gapless Bogoliubov quasiparticles with Dirac dispersion along the nodal (diagonal) directions. For our purposes, the single-particle tunneling between the $d$-wave superconductor and the QSH needs to be suppressed. This can be achieved by taking advantage of the fact that the single-particle tunneling and superconducting proximity effect have distinct spatial profiles: the former effect is peaked at the nodal direction and vanishes along the $x$ and $y$ directions, while for the latter it is the opposite. Thus, single-particle tunneling can be effectively suppressed by geometrically separating the diagonal portion QSH edge with the $d$-wave superconductor. We depict such a setup in Fig. 1, in which the corner region of the $d$-wave SC is rounded and spatially separated from the QSH layer. Alternatively, we note that nodeless $d$-wave superconductivity have been proposed for the high-$T_c$ monolayer superconductor FeSe/SrTiO$_3$ [75,76]. In addition, there are several proposals for corner pairs of MZMs with $s$-wave pairing [77,78]. All of these are free from the issue of single-particle tunneling.

For specific lattice models such a phase can be classified as a topological crystalline superconductor with time-reversal and mirror reflection symmetries, which we analyze in Appendix A by applying recent results on higher-order topological phases [39]. However, as we will see below, our analysis actually does not rely on these symmetries, and it is more general to start with a low-energy theory describing the edge modes of a QSH, which we do below. For a full lattice model and its higher-order topology, we refer the reader to Appendix A.

The existence of the corner Majorana pairs can be demonstrated by analyzing the boundary states of the QSH and treating a superconducting gap $\Delta$ and a Zeeman field $\mathbf{B}$ as perturbations. Consider a portion of the edge near a corner of a QSH insulator shown in Fig. 1. A low-energy field theory model of the QSH edge consists of a right-moving fermion $R(s)$ with spin up (in the $z$-direction) and a left-moving fermion $L(s)$ with spin down (also in the $z$-direction). These operators have standard anticommutation relations, for example $\{R(s), R^\dagger(s')\} = \delta(s-s')$, and they also obey periodic boundary conditions. The kinetic energy for this system takes the

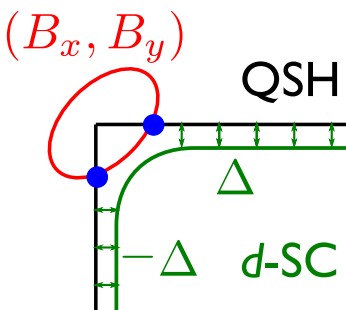

Figure 1: The schematics of the proposed setup between a QSH insulator and a $d$-wave superconductor coupled via superconducting proximity effect. The corner of the sample is subject to a Zeeman field and hosts two MZMs. The thick green curve denotes the boundary of a $d$-wave superconductor, and the interior of the red curve denotes the Zeeman field region.

low-energy form

$$H_0 = -i \int ds \left( R^\dagger(s)\partial_s R(s) - L^\dagger(s)\partial_s L(s) \right) . \tag{1}$$

Here we have assumed that both fermions have the same velocity which is set to 1. (Note that, throughout this work, we will use $s$ and $s'$ for coordinates along the edge of the 2D sample.)

The superconducting gap term, being a spin-singlet one, takes the low-energy form on the edge

$$H_{\text{SC}} = \int ds \left[ \Delta(s)R(s)L(s) + \text{h.c.} \right] . \tag{2}$$

Importantly, due to the $d$-wave pairing symmetry, the gap function is odd under mirror reflection, and when projected onto the edge, $\Delta(s)$ is an odd function (we choose the origin at the corner). Finally, the Zeeman term is projected to the edge as

$$H_{\text{Z}} = \int ds \left[ B R^\dagger(s)L(s) + \text{h.c.} \right] , \tag{3}$$

where $B \equiv B_x + iB_y = |B|e^{2i\tau}$. The full Hamiltonian is the sum of all three of these terms, $H = H_0 + H_{\text{SC}} + H_{\text{Z}}$.

This Hamiltonian can be diagonalized in the standard way by constructing lowering operators of the form $\mathcal{O}_\eta = \int dx \left\{ \eta_1(s)R(s) + \eta_2(s)R^\dagger(s) + \eta_3(s)L(s) + \eta_4(s)L^\dagger(s) \right\}$ that satisfy $[H, \mathcal{O}_\eta] = -E\mathcal{O}_\eta$ with a non-negative energy $E \geq 0$. Indeed, imposing this relation leads to the usual Bogoliubov-de Gennes equations for the "spinor"
$\eta(s) = (\eta_1(s), \eta_2(s), \eta_3(s), \eta_4(s))^T$. Without loss of generality, taking $\Delta$ and $B$ as real (their constant phases can be absorbed into the definition of $R$ and $L$), we get

$$(i\Gamma_1 \partial_s + \Delta(s)\Gamma_2 + B\Gamma_{13}) \eta(s) = E\eta(s) , \tag{4}$$

where $\Gamma_1, \Gamma_2$, and $\Gamma_{13}$ are $4 \times 4$ matrices defined as

$$\Gamma_1 = s_z \otimes \mathbb{I} , \tag{5a}$$
$$\Gamma_2 = s_y \otimes \tau_y , \tag{5b}$$
$$\Gamma_{13} = s_x \otimes \tau_z , \tag{5c}$$

and where $s_{x,y,z}$ and $\tau_{x,y,z}$ are the Pauli matrices. We note that $\{\Gamma_1, \Gamma_2\} = \{\Gamma_1, \Gamma_{13}\} = 0$, but $[\Gamma_2, \Gamma_{13}] = 0$. This means that the superconducting and ferromagnetic mass terms compete with each other.

To analyze this system we can use the fact that $[\Gamma_2, \Gamma_{13}] = 0$ to rotate to a basis in which $\Gamma_2$ and $\Gamma_{13}$ are both diagonal. The required unitary matrix $U$ is given by

$$U = \frac{1}{2} \begin{pmatrix} -1 & -1 & -1 & 1 \\ -1 & 1 & 1 & 1 \\ -1 & -1 & 1 & -1 \\ 1 & -1 & 1 & 1 \end{pmatrix}. \tag{6}$$

If we define a new spinor $\tilde{\eta}(s)$ via $\eta(s) = U^\dagger \tilde{\eta}(s)$, then we find that $\tilde{\eta}(s)$ satisfies

$$\left( i\tilde{\Gamma}_1 \partial_s + \Delta(s)\tilde{\Gamma}_2 + B\tilde{\Gamma}_{13} \right) \tilde{\eta}(s) = E\tilde{\eta}(s), \tag{7}$$

with

$$\tilde{\Gamma}_1 = s_x \otimes \tau_z, \tag{8a}$$
$$\tilde{\Gamma}_2 = s_z \otimes \mathbb{I}, \tag{8b}$$
$$\tilde{\Gamma}_{13} = s_z \otimes \tau_z. \tag{8c}$$

The key property of this new equation for $\tilde{\eta}(s)$ is that it breaks up into two decoupled $2 \times 2$ Dirac equations with masses equal to $M_\pm(s) = \Delta(s) \pm B$. Just as in the Jackiw-Rebbi model, a fermion zero mode is associated with each domain wall in $M_+(s)$ and $M_-(s)$, i.e., where $\Delta(s) = B$ or $\Delta(s) = -B$ (see Fig. 2). For the profile given by solid lines in Fig. 2, there is a mass domain wall in $M_-(s)$ (marked by the blue dot to the left), and the zero-energy solution is

$$\eta(s) = \frac{1}{2} \begin{pmatrix} i \\ -i \\ 1 \\ 1 \end{pmatrix} e^{-\int_{s_0}^{s} ds' |M_-(s')|}, \tag{9}$$

where $s_0$ is the location where $\Delta = B$. Another MZM, located at $M_+(s) = 0$, marked by the blue dot to the right in Fig. 2, can be similarly obtained. It is straightforward to verify that both solutions for $\mathcal{O}_\eta$ are Hermitian, and correspond to Majorana fermions. Therefore we find that, for odd $\Delta(s)$, there exist a pair of MZMs separated by a length $\ell$, the length of the region where $|\Delta| < B$. If $B = 0$, the two MZMs overlap in space, and form a Dirac zero mode. Indeed, they form a Kramers doublet required by the time-reversal symmetry.

Interestingly, we note that as one tunes the Zeeman field in a given direction through zero, the two MZMs swap positions. We can also consider an alternative configuration in which $\Delta$ is a constant and $B(s)$ changes sign. This is relevant for a given edge of the QSH with opposite Zeeman field applied to the two corners it connects, which we will discuss in Sec 4.2. By the same token, MZMs are nucleated at the nodes of $\Delta \pm B(s) = 0$. These two MZMs switch positions when $\Delta$ is tuned through zero. Later we will build upon this observation and propose a protocol for non-Abelian braiding of the Majorana modes.

## 2.1 Emergent symmetries

As we discuss in Appendix A in a specific lattice model, the corner Majorana modes are protected by crystalline (mirror) symmetries of the bulk theory. However, these symmetries are rather restrictive for manipulating of the Majoranas, and in experimental realizations of QSH these symmetries may not be present anyway. Here we show that fortunately there are several *emergent* symmetries in the edge theory that protect the MZMs and allow for additional perturbations to be included.

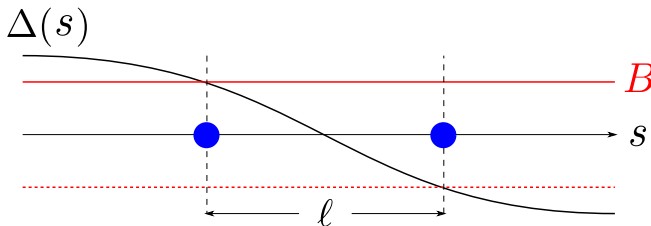

Figure 2: A spatially varying superconducting mass $\Delta(s)$ (blue curve) and a constant magnetic field $B$ (solid red line). The dashed red line is the curve $-B$. For a QSH edge with this configuration of $\Delta(s)$ and $B$, single Majorana fermions (represented by the blue dots) are localized at the points where $\Delta(s) = B$ and where $\Delta(s) = -B$. The central region where $B \geq |\Delta(s)|$ has a length $\ell$.

The edge theory including both the pairing field $\Delta$ and a Zeeman field $\mathbf{B}$ can be written in first quantized BdG form as

$$
\begin{aligned}
H =& k s_z \tau_0 + \text{Re}\,\Delta(s) s_y \tau_y + \text{Im}\,\Delta(s) s_y \tau_x \\
& + B_x s_x \tau_z + B_y s_y \tau_0,
\end{aligned}
\tag{10}
$$

where, e.g. $s_z \tau_0 := s_z \otimes \tau_0$. Here $s_z = \pm 1$ distinguishes two counter-propagating modes, which transform and couple to external field like physical spin, which we will refer to as such. We define $\mathbf{B} = (B_x, B_y)$ coupling to edge modes in such a way as the "in-plane" fields, while it is understood that they may not lie in the plane of the two-dimensional system.

First we give the full emergent symmetries when $\Delta$ and $B$ are absent. There are two U(1) subgroups: $\text{U}(1)_s$ generated by spin:

$$
U_{s,\phi} = \exp(i s_z \tau_z \phi / 2),
\tag{11}
$$

and $\text{U}(1)_c$ generated by charge

$$
U_{c,\theta} = \exp(i \tau_z \theta / 2).
\tag{12}
$$

Note that $U_{c,2\pi} = U_{s,2\pi} = -s_0 \tau_0$ is the fermion parity symmetry. The theory also enjoys a number of discrete symmetries. It is invariant under the time-reversal symmetry

$$
\mathcal{T} = i s_y K, \quad \mathcal{T} H(k) \mathcal{T}^{-1} = H(-k),
\tag{13}
$$

where $K$ is the complex conjugation. In addition, there is a chiral symmetry

$$
\mathcal{C} = s_y, \quad \mathcal{C} H(k) \mathcal{C}^{-1} = -H(k).
\tag{14}
$$

As we will see in the next Section, $\mathcal{C}$ is an anti-unitary charge-conjugation symmetry for many-body states. [79] The BdG Hamiltonian (10) also has a particle-hole symmetry

$$
\mathcal{P} = \tau_x K, \quad \mathcal{P} H(k) \mathcal{P}^{-1} = -H(-k),
\tag{15}
$$

which is not a physical symmetry, but rather a redundancy of the BdG formalism.

Now we consider the full Hamiltonian (10). The in-plane Zeeman field $B_{x,y}$ breaks the spin rotation and the time-reversal symmetries, but is invariant under the composite symmetry

$$
\tilde{\mathcal{T}} \equiv U_{s,\pi} \mathcal{T} = s_x \tau_z K,
\tag{16}
$$

which is a symmetry of (10) for a uniform phase of $\Delta$ (taken to be real without loss of generality). Such an anti-unitary symmetry which squares to one, along with $\mathcal{C}$ places the edge

theory in class BDI, which admits a $\mathbb{Z}$ classification, corresponding to a winding number that equals the number of symmetry protected MZMs. Therefore, the two MZMs at a given corner can be viewed as being protected by $\tilde{\mathcal{T}}$. In addition, we note that $U_{s,2\pi} = -s_0\tau_0$, the fermion parity, is obviously still a symmetry.

In the opposite situation in which the magnitude of the unidirectional Zeeman field (say $\mathbf{B} = B_x\hat{x}$) is spatial dependent and has a domain wall and $\max(B) > |\Delta|$, two of the Majorana modes from different corners move to the edge connecting the two corners. In this case, the composite symmetry $\tilde{\mathcal{T}}$ does not protect MZMs from different corners from hybridizing (their winding numbers under BDI are opposite), unless additional symmetries exist. Similar to the spin symmetry, the pairing field breaks both U(1)$_c$ and the chiral symmetry, but preserves their combination

$$\tilde{\mathcal{C}} \equiv U_{c,\pi}\mathcal{C} = s_y\tau_z. \tag{17}$$

With the composite chiral symmetry $\tilde{\mathcal{C}}$, the edge theory additionally belongs to class AIII, which admits another $\mathbb{Z}$ classification. Such a classification protects the two corner Majorana modes overlapping on the edge – as can be verified from Eq. (9) and its counterpart for the other corner, they carry opposite quantum numbers of the unitary operator $\tilde{\mathcal{C}}\mathcal{P}\tilde{\mathcal{T}} = s_z\tau_x$.

So far we have identified the symmetries at the level of the effective BdG Hamiltonian for the edge states. Typically some of the symmetries are not exact in the microscopic theory. For instance, in the bulk theory of the QSH, in general due to the Rashba-type spin-orbit coupling, the spin of edge states $L(s)$ and $R(s)$ may depend on momentum and on location of the edge. However, U(1)$_s$ is realized as an emergent symmetry at low energies as long as the pairing gap is much greater than the specific spin-orbit coupling that causes momentum and position dependent spin texture. For the quantum spin Hall material WTe$_2$, however, we note that recent theoretical [64] and experimental works [65] have shown that indeed there exists a spin axis for the edge states and a U(1)$_s$ symmetry at the microscopic level.

Similarly, while $\mathcal{C}$ is an exact symmetry of our lattice model for QSH in Appendix A, a generic QSH insulator is not particle-hole symmetric. However, for the edge theory, $\mathcal{C}$ emerges as an approximate symmetry as long as the chemical potential is tuned to the crossing point of the helical edge states. For a 2d system, this can be experimentally achieved via gating.

Finally, we note that while our analysis of the emergent symmetries for a given corner and for a given edge appear quite different within the BdG formalism, as we shall see, within the field-theoretical approach, the treatments for a given corner and for a given edge are completely symmetric. In fact, the T-duality of the compact free boson theory, the (1+1)d version particle-vortex duality [80], relates the two symmetries $\tilde{\mathcal{C}}$ and $\tilde{\mathcal{T}}$.

## 3  Corner MZMs from bosonization

We now switch to a bosonization description of the edge of a QSH insulator and the resulting ground state degeneracy representing the MZMs in the presence of a proximity SC field. The bosonized treatment has two advantages. First, interactions can be easily incorporated by turning on a Luttinger parameter $K \neq 1$ and by generalizing to a fractional QSH (FQSH) state. In particular, we obtain the scaling behavior of localization length of the Majorana zero modes upon varying the Luttinger parameter. Second, the calculation of the non-Abelian Berry phases are rather transparent in the bosonized formalism, which has an analog of the Berry phases in a 1d lattice.

For the sake of generality, in the bosonized theory we replace the QSH insulator with a $\nu = 1/m$ fractional quantum spin Hall (FQSH) state [81,82] and include a Luttinger parameter $K$ to capture interaction effects. The non-interacting QSH state we have focused on thus far

corresponds to the special case with $m = 1$ and $K = 1$. We note that in a recent work [83], the authors developed a similar bosonization apporach to Majorana zero modes for a non-interacting open system.

### 3.1 Review of bosonization

The edge of the FQSH state with an emergent $U(1)_s$ symmetry can be described by two bosonic fields $\phi_\uparrow(s)$ and $\phi_\downarrow(s)$ obeying the commutation relations

$$\left[\phi_\uparrow(s), \partial_{s'}\phi_\uparrow(s')\right] = \frac{2\pi i}{m}\delta(s - s'), \tag{18a}$$

$$\left[\phi_\downarrow(s), \partial_{s'}\phi_\downarrow(s')\right] = -\frac{2\pi i}{m}\delta(s - s'), \tag{18b}$$

$$\left[\phi_\uparrow(s), \phi_\downarrow(s')\right] = 0. \tag{18c}$$

In the K-matrix formalism for edges of Chern-Simons theories, this system corresponds to the matrix $K = m\sigma^z$. Both fields $\phi_{\uparrow/\downarrow}(s)$ are defined to have compactification radius $2\pi$. This means that all physical operators must be invariant under the shift $\phi_\uparrow(s) \to \phi_\uparrow(s) + 2\pi$, and likewise for $\phi_\downarrow(s)$. Then the allowed operators containing zero derivatives of these fields must be built from exponentials of the form $e^{in\phi_{\uparrow/\downarrow}(s)}$ for some integer $n \in \mathbb{Z}$.

The charge density current, and spin operator for the edge are defined to be

$$\rho(s) = \frac{1}{2\pi}\left(\partial_s\phi_\uparrow(s) + \partial_s\phi_\downarrow(s)\right),$$

$$j(s) = \frac{1}{2\pi}\left(\partial_s\phi_\uparrow(s) - \partial_s\phi_\downarrow(s)\right),$$

$$S(s) = \frac{1}{4\pi}\left(\partial_s\phi_\uparrow(s) - \partial_s\phi_\downarrow(s)\right). \tag{19}$$

We then find that the right- and left-moving electron operators for the free fermion case are given by

$$R(s) \sim \frac{1}{\sqrt{\ell}} : e^{-im\phi_\uparrow(s)} :, \tag{20a}$$

$$L(s) \sim \frac{1}{\sqrt{\ell}} : e^{im\phi_\downarrow(s)} :, \tag{20b}$$

where $\ell$ is the length of the system. We present the details of the bosonization dictionary in Appendices B and C for $m = 1$ and in Appendix F for $m \neq 1$.

With this definition we find that acting with $R(s)$ or $L(s)$ lowers the total charge by one unit, as expected for an operator that annihilates a single electron. In addition, the anticommutation relation $\{R(s), R(s')\} = 0$ (which should be obeyed by any fermionic operator) follows from the fact that $m$ is an odd integer.

The basic kinetic energy term for the bosonic fields $\phi_{\uparrow/\downarrow}(s)$ takes the form

$$H_0 = \frac{1}{2\pi}\int ds \left[\frac{1}{2}\sum_{\sigma=\uparrow,\downarrow}(\partial_s\phi_\sigma(s))^2 + g\partial_s\phi_\uparrow(s)\partial_s\phi_\downarrow(s)\right]. \tag{21}$$

Here we have also incorporated a density-density interaction (with coupling constant $g$) between the spin up and spin down fermions.[1] We see that $g > 0$ corresponds to a repulsive interaction, while $g < 0$ corresponds to an attractive interaction.

---

[1]Note that $\frac{1}{2\pi}\partial_s\phi_\sigma(s)$ is the density operator for excitations with spin $\sigma \in \{\uparrow, \downarrow\}$.

For the domain wall configurations that we study in this paper, it is convenient to introduce new non-chiral fields $\varphi(s)$ and $\vartheta(s)$ defined as

$$\varphi(s) = \frac{m}{2}\big(\phi_\uparrow(s) + \phi_\downarrow(s)\big)\,, \tag{22a}$$

$$\vartheta(s) = \frac{1}{2}\big(\phi_\uparrow(s) - \phi_\downarrow(s)\big)\,, \tag{22b}$$

which satisfy the commutation relation

$$[\varphi(s), \partial_{x'}\vartheta(s')] = \pi i\delta(s - s')\,. \tag{23}$$

In terms of these fields we find that

$$\rho(s) = \frac{1}{\pi m}\partial_s\varphi(s)\,,$$

$$j(s) = \frac{1}{\pi}\partial_s\vartheta(s)\,,$$

$$S(s) = \frac{1}{2\pi}\partial_s\vartheta(s), \tag{24}$$

and that $H_0$ can be rewritten in the form

$$H_0 = \frac{v}{2\pi}\int ds\left[\frac{1}{mK}(\partial_s\varphi(s))^2 + mK(\partial_s\vartheta(s))^2\right]\,, \tag{25}$$

where the renormalized velocity $v$ and Luttinger parameter $K$ are related to the coupling constant $g$ as

$$v = \frac{1}{m}\sqrt{1 - g^2}\,, \tag{26a}$$

$$K = \sqrt{\frac{1 - g}{1 + g}}\,. \tag{26b}$$

Note the singularity in $K$ at $g = -1$ and the zeros in $v$ and $K$ at $g = 1$. In addition, we have $K < 1$ for repulsive interactions and $K > 1$ for attractive interactions, while $K = 1$ in the absence of interactions ($g = 0$). For later use it is convenient to combine $m$ and $K$ into a modified Luttinger parameter

$$K' = mK\,, \tag{27}$$

as it is this modified Luttinger parameter that actually appears in $H_0$.

In this bosonized formalism, a superconducting mass term takes the form

$$\Delta R^\dagger(s)L^\dagger(s) + \text{h.c.} \propto \cos\big[2m\vartheta(s) + 2\rho\big]\,, \tag{28}$$

where $\rho$ is the superconducting phase. Similarly, a ferromagnetic mass term takes the form

$$(B_x + iB_y)R^\dagger(s)L(s) + \text{h.c.} \propto \cos\big[2\varphi(s) + 2\tau\big], \tag{29}$$

where $B_x + iB_y = |B|e^{2i\tau}$. A more rigorous derivation of the mass terms in terms of boson fields is done using the mode expansion, as we describe below and in Appendix D.

It is instructive to see how the bosonic variables $\varphi$ and $\vartheta$ transform under the emergent symmetries identified in the previous section:

$$\begin{aligned} U_{s,\phi} &: \vartheta \to \vartheta,\ \varphi \to \varphi - \phi/2\,, \\ U_{c,\theta} &: \varphi \to \varphi,\ \vartheta \to \vartheta - \theta/2\,, \\ \mathcal{T} &: \vartheta \to -\vartheta,\ \varphi \to \varphi + \pi/2\,, \\ \mathcal{C} &: \varphi \to -\varphi,\ \vartheta \to \vartheta - \pi/2\,. \end{aligned} \tag{30}$$

Despite their different forms for the BdG Hamiltonian, at the field theory level both $\mathcal{C}$ and $\mathcal{T}$ are antiunitary symmetries, since each flips the sign on one of the dual fields $\vartheta$ and $\varphi$. In particular $\mathcal{C}$ flips the sign of the charge but not the current, which is thus an anti-unitary charge conjugation symmetry.

The composite symmetries $\tilde{\mathcal{T}} = U_{s,\pi}\mathcal{T}$ and $\tilde{\mathcal{C}} = U_{c,\pi}\mathcal{C}$ now become

$$\begin{aligned} \tilde{\mathcal{T}} &: \ \vartheta \to -\vartheta, \ \varphi \to \varphi, \\ \tilde{\mathcal{C}} &: \ \varphi \to -\varphi, \ \vartheta \to \vartheta. \end{aligned} \tag{31}$$

Thus under $\tilde{\mathcal{T}}$ ($\tilde{\mathcal{C}}$), the Zeeman (SC) term remains invariant. Under the T-duality of the free boson theory $\varphi \longleftrightarrow \vartheta$ (which is an emergent symmetry when $K = 1$), $\tilde{\mathcal{T}}$ and $\tilde{\mathcal{C}}$ are exchanged, as well as the Zeeman and the SC terms.

In our analysis below we will mainly invoke $U_{s,\phi}$, $\tilde{\mathcal{T}}$ for the MZMs localized at a given corner, and due to the T-duality, the results directly carry over to the case of MZMs overlapping on an edge.

## 3.2 Derivation of the corner modes

We analyze the states hosted by a corner region with Zeeman fields sandwiched between two superconducting regions with opposite pairing gap related by the $d$-wave symmetry. To simplify the calculations, let us fix the length of the magnetic region to be $\ell$, within which the Zeeman field is a constant, and take the limit in which the superconducting gap $|\Delta| \to \infty$ in the superconducting region and thus the $\vartheta$ at the two ends of the magnetic region are completely pinned. Without loss of generality, we take

$$\vartheta(0) = 0 \mod \frac{\pi}{m}, \quad \vartheta(\ell) = \frac{\pi}{2m} \mod \frac{\pi}{m}. \tag{32}$$

In the magnetic region, the Hamiltonian is given by

$$H = \int_0^\ell ds \left\{ \frac{v}{2\pi}\left[ \frac{1}{mK}(\partial_s\varphi(s))^2 + mK(\partial_s\vartheta(s))^2 \right] + b\cos[2\varphi(s) + 2\tau] \right\},$$

where $b$ is a coupling constant induced by $B$. To analyze the low-energy spectrum of this Hamiltonian it is helpful to perform a mode expansions for $\varphi$ and $\vartheta$:

$$\varphi(s) = mq - \sum_{n=1}^\infty \frac{e^{-\frac{\epsilon n}{2}}}{\sqrt{n}} \cos(\kappa_n x)(b_n + b_n^\dagger), \tag{33a}$$

$$\vartheta(s) = \left(p + \frac{1}{2}\right)\frac{\pi x}{m\ell} + i\sum_{n=1}^\infty \frac{e^{-\frac{\epsilon n}{2}}}{\sqrt{n}} \sin(\kappa_n x)(b_n - b_n^\dagger),$$

where $\kappa_n = \frac{\pi n}{\ell}$, and where we included the dimensionless ultraviolet cutoff $\epsilon$ to control the oscillator sums. Removing the cutoff corresponds to taking $\epsilon \to 0$, and one can check that in this limit the fields obey the correct commutation relations in Eq. (23) (see Appendix C.3). One can also see that the field $\vartheta(s)$ obeys the boundary conditions from Eq. (32) with quantized winding number $p \in \mathbb{Z}$. Importantly, from Eq. (23) we have the commutation relation

$$[q,p] = i, \ [b_n, b_n^\dagger] = 1, \tag{34}$$

indicating $q$ and $p$ are conjugate variables. As a result of the quantization of $p$, $q$ is a compact variable with $q \sim q + 2\pi$.

In the absence of the Zeeman field, the eigenstates of $H_0$ are labeled by the winding number $p$ and the occupation number for a set of new quasiparticle modes:

$$H_0 = \frac{\pi v K}{2m\ell}\left(p + \frac{1}{2}\right)^2 + v\sum_n \kappa_n a_n^\dagger a_n + \text{const.} \tag{35}$$

The operators $\{a_n\}$ are related to $\{b_n\}$ via a Bogoliubov transformation

$$a_n = \cosh(\eta)b_n + \sinh(\eta)b_n^\dagger, \quad e^{-2\eta} = K, \tag{36}$$

which we discuss in details in Appendix D. The Fock space structure is guaranteed by $\tilde{\mathcal{T}}$ symmetry; for example a symmetry breaking Zeeman term $\sim B_z \partial_x \vartheta$ term would condense the quasiparticles in the ground state.

The Zeeman field term makes the quasiparticles massive, and further increases the quasiparticle gap. Therefore, for low-energy states we only need to focus on the Fock vacuum sector and consider the $q$ and $p$ modes. The effective Hamitonian is given by

$$H_{\text{eff}} = \alpha\tilde{p}^2 - \beta\cos(2mq + 2\tau), \tag{37}$$

in which $\tilde{p} \equiv p + 1/2$, and $\alpha, \beta$ are coupling constants renormalized by quasiparticle fluctuations. In Appendices D and F, using a variational approximation, we derive the coefficients for $K' < 2$, and for a strong Zeeman field within the range

$$B \gg \frac{1}{a}\left(\frac{a}{\ell}\right)^{2-K'}, \tag{38}$$

where $a$ is a short-distance cutoff, e.g., given by the underlying lattice. The results are given by

$$\alpha = \frac{vK'}{2\pi m^2\ell},$$
$$\beta \sim B^{\frac{2}{2-K'}} a^{\frac{2K'-2}{2-K'}}\ell. \tag{39}$$

Using Eq. (38), it is straightforward to see that here $\beta \gg \alpha$.[2]

The Schrödinger equation with the Hamiltonian in Eq. (37) is known as the Mathieu's equation [84–86], and can be viewed as the equation of motion of a single particle in a 1d ring modeled by a periodic lattice potential. According to Bloch's theorem, the eigenstates $|\psi_k\rangle$ of this Hamiltonian is labeled by lattice momenta $k$,[3] i.e. $e^{i\frac{\pi}{m}\tilde{p}}|\psi_k\rangle = e^{i\frac{\pi}{m}k}|\psi_k\rangle$, which take quantized values inside the Brillouin zone. The lattice constant is $\pi/m$, and thus $k \in [-m, m)$. Since $\tilde{p}$ takes half-integer quantized values, so does $k$. The offset $1/2$ in the quantization of $\tilde{p}$ is analogous to the effect of a magnetic flux through the lattice ring, causing a twisted boundary condition and the same amount of offset in the lattice momenta $k$. Therefore, we have

$$k \in (\mathbb{Z} + 1/2) \cap (-m, m). \tag{40}$$

Under $\tilde{\mathcal{T}}$, $\tilde{p}, k \to -\tilde{p}, -k$. From Eqs. (24) and (33), we see that physically the tunneling current and spin quantum numbers are directly related to the $k$ via

$$J = 2S = \int_0^\ell \frac{ds}{m\pi}\partial_x\vartheta(s) = \frac{k}{m} \quad \text{mod } \frac{1}{m}. \tag{41}$$

---

[2]Interestingly, we note that for the free fermion case with $K = 1$, the prefactor $\beta$ of the cosine term is actually proportional to $B^2$ rather than $B$.

[3]This is not to be confused with the *actual* lattice momenta of our QSH/$d$-SC system.

The eigenstates of this Hamiltonian form energy bands labeled by the band index and lattice momenta $\{k\}$. Each band consists of $2m$ states. We will focus on the lowest band.

In the limit of large $\ell$ or large $B$, we have $\beta \gg \alpha$ and the 1d lattice is in a flat-band limit. As we show in Appendix E.1, in the limit $\beta \gg \alpha$ (which is the same as Eq. (38)) the bandwidth is exponentially suppressed as

$$\Delta E \sim \text{const} \times \exp\left(-\frac{\ell}{\xi}\right). \tag{42}$$

Here the correlation length $\xi$ for $K' < 2$ is expressed as

$$\xi = \sqrt{\frac{\alpha}{\beta}}\ell \sim \left(\frac{1}{Ba}\right)^{\frac{1}{2-K'}} a, \tag{43}$$

following the familiar Kosterlitz-Thouless scaling behavior. The $2m$ states in the lowest band are approximately degenerate. This is the same $2m$ degeneracy given by a pair of $\mathbb{Z}_{2m}$ parafermions [69–71, 87–89].

In particular for $m = 1$, the ground state degeneracy corresponds to a pair of MZMs, consistent with what we found using the BdG formalism. The exponential supression of the hybridization energy of the parafermions indicates that these modes are exponentially localized in space, consistent with the results we obtained for the free fermion case. Indeed, for $m = 1, K = 1$ we restore the familiar result $\xi \sim 1/B$; see Eq. (9).

In the flat band limit $\beta \gg \alpha$, the gap separating the MZMs from excited states can be obtained by approximating Eq. (37) by expanding the cosine potential. We find for the excitation gap

$$\Delta_{\text{ex}} \sim \sqrt{\alpha\beta} = \sqrt{K}B(Ba)^{\frac{K-1}{2-K}}. \tag{44}$$

For the free fermion system, this expression reduces to the Zeeman energy $\sim B$.

In the opposite limit with $B \ll \frac{1}{a}\left(\frac{a}{\ell}\right)^{2-K'}$, the Mathieu's equation is in the weak potential limit, and the band dispersion is similar to that of a free particle. In particular, as $B$ goes to zero, the band gap closes at the BZ boundary ($\pm 1$ for $m = 1$) and the spectrum restores the parabolic dispersion. If either of the states in the lowest bands reside at the Brillouin zone boundary, the MZMs will be "poisoned" by excited states.

Fortunately, in the presence of the twist boundary condition causing $k \in \mathbb{Z} + 1/2$, the quantized lattice momentum $k$ do not take values at the BZ boundary, and the Majorana states in the lowest band remain degenerate and separated from higher bands. This is consistent with our findings in the previous Section. The size of their spatial profile is $\mathcal{O}(\ell)$. The analysis here further shows that the excitation gap results from the quantization of lattice momentum, and is given by

$$\Delta_{\text{ex}} \sim v/\ell. \tag{45}$$

We prove the two-fold degeneracy for a general $\beta/\alpha$ more rigorously in Appendix G.

Interestingly, such a robust ground state degeneracy has been elucidated [90] from the perspective of a mixed 't Hooft anomaly between time-reversal symmetry (corresponding to our generalized time-reversal $\tilde{\mathcal{T}}$) and fermion parity symmetry $\varphi \rightarrow \varphi + \pi$ (generated by our $U_{s,2\pi}$) of field theories with a $\Theta$-term at $\Theta = \pi$. The anomaly ensures that independent of basis choice, one of the two classical symmetries is represented as a double cover at the quantum level, leading to the two-fold degeneracy. The two states are related by time-reversal and differ by fermion parity quantum numbers. We present the proof of the degeneracy in Appendices H in a way that reveals a clear analogy with Appendix D of Ref. [90].

We end this section by noting that in the alternative configuration when the Zeeman fields at two different corners are antiparallel, the two MZMs can be obtained in a dual bosonized theory, related to our discussion above by $\vartheta \leftrightarrow \varphi$ and $\tilde{\mathcal{T}} \leftrightarrow \tilde{\mathcal{C}}$.

# 4 Symmetry-protected quantum gates of Majorana qubits

In this Section we focus on the $m = 1$ case where the degenerate corner states correspond to a pair of MZMs. As we showed in the fermionic language, these two MZMs are located at $\Delta(s) = \pm B$ and switch position when the Zeeman field is flipped. We now show that when the in-plane Zeeman field is *rotated* back, the full process induces an non-Abelian Berry phase that is the same as exchanging two-dimensional MZMs (or Ising anyons). Furthermore, we show that by tuning the Zeeman field as well as the superconducting order parameter, one can realize all Clifford gates and universal phase gates on the ground state qubit, protected by the emergent symmetries of the theory.

As is well-known, with a fixed fermion parity, four MZMs form a two-level system. This is realized by two adjacent corners of the higher-order topological superconductor platform. The Clifford gates consist of exchanging the Majorana modes both within the same corner and across different corners, the protocol of which we discuss below.

While we use terms such as "braiding" and "exchanging" the MZMs in what follows, it should be emphasized that they are distinct from their counterparts realized by directly moving the anyonic excitations. For example, in our "braiding" process, the physical locations of the MZMs are not changed, and in our "exchange" process, the MZM's exchange locations but they do not remain well separated during the process. More precisely, we use these terms to mean that the resulting non-Abelian Berry phase, when protected by symmetries, is identical to those from the braiding and exchanging anyons. In this sense our proposal is closely connected to holonomic quantum computing [91].

## 4.1 Manipulating Majorana corner modes via Zeeman field

### 4.1.1 Full braid via $2\pi$ rotation of the Zeeman field

Before we discuss the exchange process of the Majorana modes, let us first consider an adiabatic process involving a full $2\pi$ rotation of the in-plane Zeeman field and compute the Berry phase. In our conventions this corresponds to keeping the magnitude $B$ of the magnetic field fixed while tuning the angular parameter $\tau$ from 0 to $\pi$ in Eq. (37). We will show that this correspond to a full braid of two Majoranas at a given corner.

Let $|\psi_k(B, \tau)\rangle$ be the ground state of $H_{\text{eff}}(B, \tau)$ with lattice momentum $k$. According to Eq. (40), $k = \pm 1/2$. In other words,

$$e^{i\pi\tilde{p}}|\psi_k(B, \tau)\rangle = e^{i\pi k}|\psi_k(B, \tau)\rangle = \pm i|\psi_k(B, \tau)\rangle. \tag{46}$$

In the 1d lattice interpretation, the parameter rotation angle of the Zeeman field $2\tau$ corresponds to a displacement of the periodic potential by an amount of $\tau$, and thus the eigenstate can be expressed via a translation operator:

$$|\psi_k(B, \tau)\rangle \sim e^{-i\tau\tilde{p}}|\psi_k(B, 0)\rangle. \tag{47}$$

However, from Eq. (46), the right hand side is not single-valued upon a full $2\pi$ rotation ($\tau = \pi$) of the Zeeman field. For an unambiguous calculation of the Berry phase, one should choose the phases of the states $|\psi_k(B, \tau)\rangle$ so that these states are single-valued functions of $B$ and $\tau$ (defined modulo $\pi$) in the region of the parameter space that is of interest for the Berry phase calculation.

This issue can be addressed by adding to each ground state a c-number phase factor $e^{i\tau k}$ corresponding to their lattice momenta

$$|\psi_k(B, \tau)\rangle = e^{i\tau k}e^{-i\tau\tilde{p}}|\psi_k(B, 0)\rangle . \tag{48}$$

From Eq. (46), this choice ensures that $|\psi_k(B,\tau)\rangle$ returns to itself when we wind the Zeeman field by $2\pi$ (translating $\tau$ by $\pi$), i.e., we have

$$|\psi_k(B,\tau+\pi)\rangle = |\psi_k(B,\tau)\rangle \; . \tag{49}$$

This is not the only possible choice, and it is well-known that the Berry phases that we obtain are invariant under any redefinition $|\psi_k(B,\tau)\rangle \to |\tilde{\psi}_k(B,\tau)\rangle = e^{i\theta_k(B,\tau)}|\psi_k(B,\tau)\rangle$, provided that the new states $|\tilde{\psi}_k(B,\tau)\rangle$ are also single-valued functions of $B$ and $\tau$ in the relevant region of the parameter space.

We now compute the Berry phases $\gamma_k$ picked up by the states $|\psi_k(B,\tau)\rangle$ during the $2\pi$ rotation of the Zeeman field. The Berry phases $\gamma_k$ are given by the standard formula

$$\gamma_k = i \int_0^\pi d\tau \; \langle\psi_k(B,\tau)|\partial_\tau|\psi_k(B,\tau)\rangle \; , \tag{50}$$

and so using Eqs. (48) we find that

$$\gamma_k = \pi\langle\psi_k(B,0)|\tilde{p}|\psi_k(B,0)\rangle - \pi k \; . \tag{51}$$

Intuitively, the first term evaluates the average momentum of the Bloch states, and the second term lattice momentum. As we mentioned, depending on the depth of the periodic potential in Eq. (37) there are two important limits – the tight-binding limit ($\beta \gg \alpha$) and weak periodic potential limit ($\alpha \gg \beta$). In the first limit, Bloch states are approximately a linear superposition of bound states each at a potential minimum, while in the second limit, Bloch states are approximately plane-wave states. Therefore, heuristically we find that in former limit the average momentum $\langle\tilde{p}\rangle$ approaches zero, while in the latter $\langle\tilde{p}\rangle$ approaches the lattice momentum $k$. Thus we have in the tight-binding limit,

$$\gamma_k = -\pi k, \quad k = \pm\frac{1}{2}, \tag{52}$$

which can be understood as coming from "dragging" the Bloch state by a lattice constant. In the opposite limit, the Berry phase vanishes, i.e., the lattice potential is so weak that translating it does not induce a significant change in the wave function.

Here focus on the tight-binding limit ($\beta \gg \alpha$). This is the same condition as (38), i.e., $Ba \gg (a/\ell)^{2-K}$. In Appendix E.3 we directly compute the Berry phase for the state $|\psi_k(B,0)\rangle$ using our analysis of that state based on Mathieu's equation. There we show that the deviations of the Berry phase from the approximate result in Eq. (52) is indeed exponentially small in $\ell$. This result is topological, in the sense that the Berry phase does not depend on parameters such as $B$ and $K$ up to exponentially small corrections.

Noting that $k = \pm 1/2$, the result in (52) matches exactly the non-Abelian Berry phases accrued during a full $2\pi$ braiding of two MZMs. [92] We note that the full braiding of the Majorana modes have also been proposed in a similar higher-order topological superconductor platform in Ref. [56].

### 4.1.2 Single exchange via $\pi$ rotation and flip of Zeeman field

We now show that owing to symmetries of the system, one can also perform a *single exchange* of two Majoranas using a different adiabatic process that also involves only the external Zeeman field. To motivate this process, recall from the previous subsection that the Berry phase for a $2\pi$ rotation of the Zeeman field within the $x$-$y$ plane is equal (at large $\ell$) to the Berry phase for a full braid (double exchange) of the fractional quasiparticles localized near the ends of the FM region. In this subsection we show that this Berry phase, and the adiabatic process itself, can be

split into two equal contributions in a symmetry-protected manner, such that each contribution on its own yields the Berry phase for a single exchange of fractional quasiparticles. The Berry phase $\tilde{\gamma}_k$ for this process is then given by *half* of the value $\gamma_k$ for the full braid,

$$\tilde{\gamma}_k = -\frac{\pi k}{2}, \quad k = \pm\frac{1}{2}\ . \tag{53}$$

We find that $\tilde{\gamma}_{\pm 1/2}$ for the two ground states differ by $\pi/2$, and this is exactly the relative Berry phase expected for a single exchange of two MZMs [10].

To achieve this, we consider "half-moon" paths of the Zeeman field $\mathbf{B} = (B_x, B_y)$ denoted in Fig. 3. This path consists of a half circle from $\tau = 0$ to $\tau = \pi/2$, and a straight line with $B_y = 0$ sweeping the $\mathbf{B}$ field back to the initial configuration passing through the origin. Crucially, along the arc $\mathbf{B}$ must be in the tight-binding regime, i.e., $Ba \gg (a/\ell)^{2-K}$. As we shall see below, the precise shape of the arc does not matter, as long as it is in the flat-band limit. We denote such a contour transversed counterclockwise by $\mathcal{C}'_B$, and its image under inversion in the $B_x - B_y$ plane, transversed clockwise, by $\mathcal{C}_B$".

Let $|\psi_k(\mathbf{B})\rangle = |\psi_k(B_x, B_y)\rangle$ be the ground state of the Hamiltonian in the sector with "lattice momentum" $k$. In our previous notation we had $B_x = B\cos(2\tau)$ and $B_y = B\sin(2\tau)$, and so $|\psi_k(\mathbf{B})\rangle$ can be identified with the state $|\psi_k(B, \tau)\rangle$ that we defined in Eq. (48). The Berry phase $\gamma_k$ for the full $2\pi$ rotation of $\mathbf{B}$ can be written as the line integral

$$\gamma_k = i \oint_{\mathcal{C}_B} d\mathbf{B} \cdot \langle \psi_k(\mathbf{B})|\nabla_{\mathbf{B}}|\psi_k(\mathbf{B})\rangle \ , \tag{54}$$

where $\mathcal{C}_B$ is the circular contour of radius $B$ centered at the origin of the $B_x$-$B_y$ plane.

The integral expression for $\gamma_k$ can be split into two contributions as

$$\gamma_k = i \oint_{\mathcal{C}'_B} d\mathbf{B} \cdot \langle \psi_k(\mathbf{B})|\nabla_{\mathbf{B}}|\psi_k(\mathbf{B})\rangle + i \oint_{\mathcal{C}''_B} d\mathbf{B} \cdot \langle \psi_k(\mathbf{B})|\nabla_{\mathbf{B}}|\psi_k(\mathbf{B})\rangle$$
$$\equiv \gamma'_k + \gamma''_k. \tag{55}$$

Here $\gamma'_k$ and $\gamma_k$" are the contributions to the total Berry phase from the half-moon paths $\mathcal{C}'_B$ and $\mathcal{C}''_B$, respectively.

An important prerequisite for a well-defined Berry phase is that the system remains gapped during the process. This is indeed true for the half-moon contour, since the ground state qubit is always energetically separated from the excited states: on the outer arc, the corner region is gapped by the Zeeman field (see Eq. (44)). Near the origin, the Zeeman field vanishes but a finite size gap still exists (see Eq. (45)). In addition, in order for the dynamical phases to cancel, the two ground states should remain degenerate, which is guaranteed by the $\tilde{\mathcal{T}}$ symmetry for any value of $B$, including at $B = 0$.

Provided that the Zeeman field is the only odd component under the $\mathcal{T}$ symmetry (the physical time-reversal symmetry preserved by the quantum spin Hall and $d$-wave superconductor; not to be confused with $\tilde{\mathcal{T}}$), the contour $\mathcal{C}'_B \to -\mathcal{C}''_B$ under $\mathcal{T}$ (which reverses both the Zeeman field and the orientation of the contour). The Berry phase is obviously odd under time-reversal, and therefore,

$$\gamma'_k - \gamma_k" = i \oint_{\mathcal{C}'_B} d\mathbf{B} \cdot \langle \psi_k(\mathbf{B})|\nabla_{\mathbf{B}}|\psi_k(\mathbf{B})\rangle + i \oint_{-\mathcal{C}''_B} d\mathbf{B} \cdot \langle \psi_k(\mathbf{B})|\nabla_{\mathbf{B}}|\psi_k(\mathbf{B})\rangle$$
$$= 0. \tag{56}$$

Combining Eqs. (55, 56), we see that

$$\gamma'_k = \gamma_k" = \tilde{\gamma}_k = -\frac{\pi k}{2}, \quad k = \pm\frac{1}{2}, \tag{57}$$

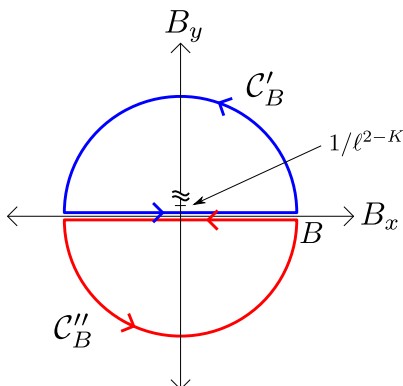

Figure 3: The "half-moon"-shaped contours $\mathcal{C}'_B$ (blue) and $\mathcal{C}"_B$ (red) in the $B_x$-$B_y$ plane. The magnetic field in the semicircle portion is in the tight-binding regime $Ba \gg (a/\ell)^{2-K}$. The Berry phase for either one of these paths is equal (up to exponentially small corrections) to the known Berry phase for a single braid of fractional quasiparticles.

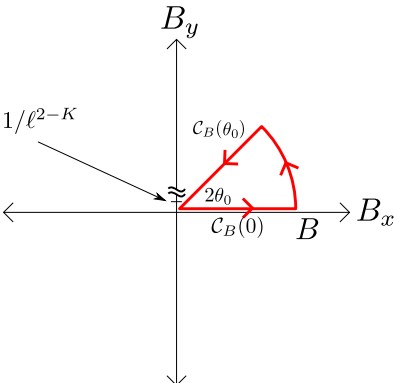

Figure 4: The "slice-of-pie" countour in the $B_x$-$B_y$ plane, which achieves a phase gate for the two Majorana modes at a given corner. The magnetic field in the semicircle portion is in the tight-binding regime $Ba \gg (a/\ell)^{2-K}$.

precisely the Berry phase during an exchange process of the Majoranas.

Let us summarize the conditions required to have the quantized value of the Berry phase:

1. The Berry phase is robust against small deformations of the arc as long as the flat-band condition is maintained.

2. The track of the magnetic field must be invariant under $\mathbf{B} \to -\mathbf{B}$ in regions other than $Ba \gg (a/\ell)^{2-K}$ and $Ba \ll (a/\ell)^{2-K}$, such as parts of the straight line segment in Fig. 3.

3. The system must have the $\mathcal{T}$ symmetry in the absence of the Zeeman field, and the $\tilde{\mathcal{T}} = U_{s,\pi}\mathcal{T}$ symmetry in its presence. This means that the phase difference between the two superconducting regions right outside the corner region must be $\pi$, which is naturally realized in our setup with a $d$-wave superconductor.

### 4.1.3 Phase gate via generic rotation of Zeeman field

In this last subsection we build on the idea of the previous subsection and show that it is possible to obtain a continuous family of Berry phase values by taking the system along a "slice of pie" path in the parameter space of the external magnetic field $\mathbf{B} = (B_x, B_y)$, shown in Fig. 4. The specific path that we consider is as follows. We start with $B_x = B > 0$ and $B_y = 0$. In the first part of the path we rotate the Zeeman field counterclockwise by an angle of $2\theta_0$, ending up at $B_x = B\cos(2\theta_0)$ and $B_y = B\sin(2\theta_0)$. In the second part of the path we traverse the straight line segment from $\mathbf{B} = (B\cos(2\theta_0), B\sin(2\theta_0))$ to the origin $\mathbf{B} = (0, 0)$. Finally, in the third part of the path we traverse the straight segment from the origin $\mathbf{B} = (0, 0)$ back to our starting point $\mathbf{B} = (B, 0)$. Crucially, we assume that the first part of the path, namely the curved segment that is traversed at constant magnitude $|\mathbf{B}| = B$, is taken in the tight-binding regime. We will also assume that in the absence of $\mathbf{B}$, the theory has U(1)$_s$ symmetry.

To calculate the Berry phase $\gamma_k(\theta_0)$ in this process, we first consider the contribution from the two straight line paths, $\mathcal{C}_B(0)$ and $\mathcal{C}_B(\theta_0)$. Since the only term in the system that violates the spin rotation symmetry $U_{s,\theta}$ in Eq. (11) is the Zeeman field, and that the two paths $\mathcal{C}_B(0)$ and $-\mathcal{C}_B(\theta_0)$ are related by the U(1)$_s$ symmetry, the total contribution to the Berry phase along the straight paths $\mathcal{C}_B(0) + \mathcal{C}_B(\theta_0)$ is zero.

Then $\gamma_k(\theta_0)$ is exactly equal to the contribution from the curved part of the path, and so

$$\gamma_k(\theta_0) = i \int_0^{\theta_0} d\tau \, \langle \psi_k(B, \tau) | \partial_\tau | \psi_k(B, \tau) \rangle \,. \tag{58}$$

If we evaluate this using our variational approximation in the large $B$ regime (following the ideas from earlier in this section), then we find the total Berry phase as

$$\gamma_k(\theta_0) = -k\theta_0, \quad k = \pm\frac{1}{2}. \tag{59}$$

We note that this argument can also be applied to the symmetry protection for the exchange process. The result (59) is protected by the spin-rotation symmetry U(1)$_s$ when there is no Zeeman field.

Again let us summarize the conditions required to have the quantized value of the Berry phase:

1. The Berry phase is robust to small deformations of the arc as long as the flat-band condition is maintained.

2. In regions outside $Ba \gg (a/\ell)^{2-K}$ or $Ba \ll (a/\ell)^{2-K}$, The tracks of magnetic field must be precisely related by a $2\theta_0$ rotation in the $B_x - B_y$ plane.

3. The system must have U(1)$_s$ symmetry when there is no Zeeman field, and the $\tilde{\mathcal{T}} = U_{s,\pi}\mathcal{T}$ symmetry in its presence to protect the ground state degeneracy. Here it is achieved by applying the Zeeman field in the direction perpendicular to the spin axis.

Due to the ground state degeneracy it is also possible to choose as the initial state a superposition of $k = \pm 1/2$. In the next subsection we discuss such a situation where we choose a different basis for initial states.

### 4.2 Clifford and phase gates via manipulating Majorana modes within and across corners

In this Subsection we consider a configuration of two adjacent corners subject to antiparallel Zeeman fields and the edge between them are gapped by SC order, which we depict in Fig. 5.

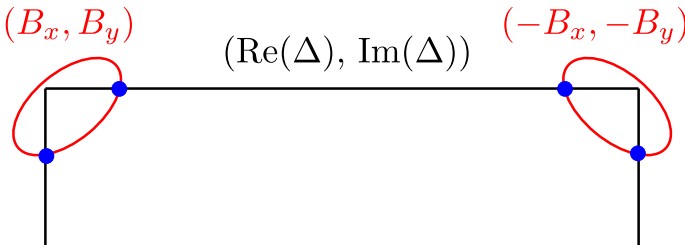

Figure 5: A qubit formed by four MZMs from adjacent corners. By tuning the Zeeman field **B** and the superconducting order $\Delta$ one can achieve symmetry protected Clifford gates.

From Eq. (24), the tunneling current through each corner is given by Eq. (41):

$$J = \int_0^\ell \frac{ds}{\pi} \partial_x \vartheta(s) = \frac{k}{m} \mod \frac{1}{m}, \tag{60}$$

corresponding to the fermion parity at the corner

$$(-)^F \equiv e^{i\pi k}. \tag{61}$$

Since quasiparticles can tunnel between corners through the edge, only the combined fermion parity of the two corners is conserved. For a given parity (say even), such a configuration with two corners and one edge form a single qubit, with the two energy levels distinguished by the parity at a given corner, which we label as

$$| \uparrow \rangle \equiv | k = \frac{1}{2}, k' = \frac{1}{2} \rangle,$$
$$| \downarrow \rangle \equiv | k = -\frac{1}{2}, k' = -\frac{1}{2} \rangle, \tag{62}$$

where $k$ and $k'$ are the respective lattice momenta in the two corners. With this notation, the exchange operation in either corner leads to a Berry phase represented by

$$\tilde{\gamma} = e^{i \frac{\pi \sigma_z}{4}}, \tag{63}$$

where $\sigma_z$ is the Pauli matrix in the Hilbert space of (62).

To realize Clifford gates, one additionally needs to achieve the non-Abelian unitary operator

$$\bar{\gamma} = e^{i \frac{\pi \sigma_x}{4}}. \tag{64}$$

In Ref. [10] this can be achieved by swapping different sets of Majorana pairs. Similarly, here we show that $\bar{\gamma}$ is achieved by manipulating two Majorana modes *across* different corners.

As we discussed in Sec. 2.1, with antiparallel in-plane Zeeman fields in the two corners the edge region is described by a theory dual to the one for the corner regions, with Majorana modes protected instead by $\tilde{C}$ symmetry. According to (59), such a duality is simply the usual $\vartheta \leftrightarrow \varphi$ duality in bosonization. In this basis, the two states forming the qubit are then eigenstates of

$$Q = \int_{\text{edge}} \frac{ds}{\pi} \partial_x \varphi(s), \tag{65}$$

which is the fermion parity in the superconducting edge. Within the subspace of the ground state qubit, the charge eigenstate is a superposition between the two different eigenstates for tunneling current $J$, thus we can rewrite $Q$ as (which is time-reversal invariant)

$$Q = | \uparrow \rangle \langle \downarrow | + | \downarrow \rangle \langle \uparrow | = \sigma_x, \tag{66}$$

and its eigenstates are labeled by $\langle \sigma_x \rangle = \pm 1$.

In order to induce Berry phases, we can similarly design contours in the complex plane of $\Delta$ similar to that of $B = B_x + iB_y$. Assuming that the system size is much larger than the superconducting coherence length, one can tune the pairing fields for different edges independently. Due to the $\vartheta \leftrightarrow \varphi$ duality, one can straightforwardly obtain that a "half-moon" contour leads to the non-Abelian phase given in (64). Notice here this value is topological and protected by the dual $\tilde{\mathcal{C}}$ symmetry, in which, as we showed $\mathcal{C}$ is an emergent symmetry guaranteed by properly gating the sample to charge neutrality.

In addition, it is straightforward to see that the phase gate operation can be realized for Majorana's across different corners by taking a "slice of pie" contour (analog of that in Fig. 4) of an angle $2\rho_0$ in $\Delta$, which is protected by the $U(1)_c$ symmetry. Thus we have two types of phase gates available, namely,

$$\tilde{\gamma}(\theta_0) = e^{i\theta_0 \sigma_z}, \quad \tilde{\gamma}(\rho_0) = e^{i\rho_0 \sigma_x}, \tag{67}$$

which for $\theta_0 = \rho_0 = \pi/8$ correspond to the magic gates [16].

We note that in Ref. [66] the authors pointed out that a qubit made out of a Kramers doublet of MZMs can be subject to a non-Abelian Berry phase in the presence of an adiabatic local perturbation without lifting the Kramers degeneracy, unless they carry distinct quantum numbers. In our case at $B = 0$, the Majorna zero modes overlap in space and form a Kramers pair. However, the two states associated with the corner MZMs are distinguished by their "lattice momenta" $k \in \{-1/2, 1/2\}$ and their fractional spins (see Eq. (41)) $S = \pm k/2$, and hence are protected by symmetry from local perturbations.

Finally, note here that so far our platform has only involved two edge-sharing corners of a semi-infinite sample. By simple math, a $d$-wave superconductor setup produces four such corners, corresponding to three qubits (with a fixed fermion parity for the sample). With the protocol above, one can realize a set of Clifford gates on each edge, leading to a richer set of quantum gates in the enlarged Hilbert space.

## 5 Braiding parafermion modes

In Sec. 4 we have completely focused on the $m = 1$ case, in which the ground states are MZMs. It is straightforward to generalize our full braid, exchange, and phase gates to a generic $m$. For example, via a half-moon contour in $B$, we obtain

$$\tilde{\gamma}_k = -\frac{\pi k}{2m}, \quad k = [-m, m) \cap (\mathbb{Z} + 1/2). \tag{68}$$

This result is exactly the exchange statistics of $\mathbb{Z}_{2m}$ parafermions. [69–71, 87–89]

However, unless $\beta \gg \alpha$, the $2m$ eigenstates in the lowest band are not degenerate. In the half-moon contour, this indicates that on the straight line portion through $B = 0$ of the contour, even though the $2m$ states remain separated from the other excited states, the topological Berry phase cannot be separated from a $k$-dependent dynamical phase that is non-universal. The $2m$ states come in $m$ pairs, leading to $m - 1$ independent relative dynamical phases.

For small $m$, it may be possible to eliminate dynamical phases by precisely controlling the system parameters and the duration of the exchange process such that it is a common period for all $m - 1$ modes. However, the result is not topological protected against unitary errors induced by imperfect cancelation of dynamical phases.

# 6 Conclusion

In this work we have proposed a platform for topological quantum computing based on a heterostructure between a high-$T_c$ $d$-wave superconductor and a quantum spin Hall insulator, which can be regarded as a higher-order topological superconductor. We demonstrated that, via tuning the a Zeeman field applied to the corner region and the superconducting order parameter, such a setup can realize non-Abelian Clifford gates of Majorana qubits that are protected by time-reversal and charge conjugation symmetries, as well as phase gates (including the $\pi/8$ magic gates needed for universal topological quantum computing) protected by U(1) symmetries. Within our analysis, interaction effects and generalization to a fractional quantum spin Hall states can naturally be incorporated.

In our proposed setup, the $d$-wave superconductor ensures a large critical temperature and a large critical field, making the manipulation of Majorana's via an external Zeeman field easier to realize in experiments. Recent advancements in low-dimensional materials have made the key components of the heterostructure, including $d$-wave superconductors (and its monolayer version [62]), quantum spin Hall insulators [63] and two-dimensional ferromagnets [67], readily available. Other than the specific combination of ingredients in our proposal, our theoretical analysis is based on low-energy effective field theories, which can be easily adapted to other topological materials with magnetism and superconductivity. We note that recently signatures of parafermions have been observed in a similar setup with a fractional quantum Hall insulator [74]. It will be extremely interesting to see if one can demonstrate and manipulate these non-Abelian anyons using the protocols we propose in this work.

# A Lattice model for second-order topological superconductor

In this Appendix we present a lattice model for the second-order topological superconductor given by a quantum spin Hall (QSH) insulator with a proximity effect induced $d$-wave super-conudcting ($d$-SC) gap, given by $H_0 + H_{SC} = \int d\mathbf{k}\Psi^\dagger(\mathbf{k})(\mathcal{H}_0 + \mathcal{H}_{SC})(\mathbf{k})\Psi(\mathbf{k})$, where $\Psi^\dagger = (\psi^\dagger(\mathbf{k}), \psi(-\mathbf{k}))$ and

$$
\begin{aligned}
\mathcal{H}_0(\mathbf{k}) + \mathcal{H}_{SC}(\mathbf{k}) = & \sin k_x s_z \sigma_z \tau_0 + \sin k_y s_0 \sigma_y \tau_z \\
& + (\cos k_x + \cos k_y + m - 1)s_0 \sigma_x \tau_z \\
& + \Delta \sin k_x \sin k_y s_y \sigma_0 \tau_y.
\end{aligned}
\tag{69}
$$

Here $s_{x,y,z}$ denotes the spin degree of freedom, $\sigma_{x,y,z}$ is a band index, and $\tau_{x,y,z}$ are Pauli matrices in the Nambu space. The first three terms describes the normal state, which is a quantum spin Hall insulator, and the last term is a pairing term of $d$-wave symmetry, coming from the proximity effect with a high-$T_c$ superconductor. The Hamiltonian has a time-reversal symmetry given by $\mathcal{T} = is_y K$, diagonal mirror symmetries $\mathcal{M}_a = s_x \sigma_z \tau_y$, $\mathcal{M}_b = s_y \sigma_y \tau_x$, and a particle-hole symmetry $\mathcal{P} = \tau_x K$.

Such a Hamiltonian has been studied in Refs. [60,61] as a second-order topological superconductor protected by time-reversal symmetry and a $C_4$ rotation symmetry. For our purposes, we will instead rely on the mirror reflection symmetries $\mathcal{M}_{a,b}$. Higher-order topological crystalline insulators and superconductors with mirror symmetries have been classified in Ref. [39] based on a $K$-theory analysis by Shiozaki and Sato [93]. In our case, the reflection symmetry anticommutes with both time-reversal and particle-hole conjugation. According to the terminology in Ref. [39], it belongs to symmetry class DIII$^{\mathcal{M}_{++}}$, which in terms of second-order topology admits a $\mathbb{Z}_2$ classification in 2d. In the nontrivial phase symmetric corners of the sample host a pair of MZMs that form Kramer doublet and have the *same* mirror eigenvalue.

Here the $\mathbb{Z}_2$ classification is an *intrinsic* bulk property. As such, one cannot remove the Majorana doublet by modifying the boundary termination without breaking the symmetry. For example, one can glue a 1d time-reversal invariant topological superconductor on one of the edges, upon coupling to the bulk, this gaps out the corner Majorana doublet, but this procedure necessarily violates mirror symmetry.

We also consider an in-plane Zeeman field, either applied throughout the bulk or only near the corners, given by $H_Z = \int d\mathbf{k} \Psi^\dagger(\mathbf{k}) \mathcal{H}_Z(\mathbf{k}) \Psi(\mathbf{k})$ where

$$\mathcal{H}_Z = B_x s_x \sigma_0 \tau_z + B_y s_y \sigma_0 \tau_0. \tag{70}$$

The Zeeman field breaks both $\mathcal{T}$ and $\mathcal{M}_{a,b}$, but preserves the composite symmetry $\mathcal{T}\mathcal{M}_{a,b}$. Importantly, the other Zeeman term $\sim B_z s_z \sigma_0 \tau_z$ is odd under this action and is forbidden. Together with the particle-hole symmetry $\mathcal{P} = \tau_x K$, the Hamiltonian $H = H_0 + H_Z$ preserves composite chiral (anti)symmetries $\mathcal{P}\mathcal{M}_{a,b} = s_z \sigma_z \tau_z$, which anticommutes with $\mathcal{P}$. Such a phase belongs to class $D^{\mathcal{P}\mathcal{M}_-}$, which also admits a $\mathbb{Z}_2$ classification. This indicates the MZMs in the absence of $H_Z$ remains robust, despite time-reversal symmetry being broken. However, as pointed out in Ref. [39], this $\mathbb{Z}_2$ invariant is *extrinsic*. In fact, as we show in the main text, the Majorana modes can be moved (or evem removed) by symmetric boundary perturbations.

Experimentally, a $d$-SC/QSH heterostructure can be achieved by stacking $d$-wave high-$T_c$ superconductor BSCCO and quantum spin Hall insulator WTe$_2$. Of course, depending material details and the geometry of stacking, such a heterostructure may not realize the mirror symmetries we specified above. However, as we show in the main text, the Majorana modes can be protected by other emergent on-site symmetries.

## B Mode expansion of the bosonic fields

In this appendix we explain in more detail the mode expansions for $\varphi(s)$ and $\vartheta(s)$ that we use in our analysis in this paper. To obtain the mode expansion for $\vartheta(s)$, we first identify a complete set of functions of $x \in (0, \ell)$ that also obey the boundary conditions Eq. (32), and then we expand $\vartheta(s)$ as a series in these functions with operator-valued coefficients. We then expand $\varphi(s)$ in terms of a complementary set of functions (also with operator-valued coefficients) in such a way that $\vartheta(s)$ and $\varphi(s)$ obey the correct commutation relations.

In our case the operator-valued coefficients that appear in the mode expansions consist of zero mode operators $q$ and $p$ and a set of oscillator raising and lowering operators $b_n$ and $b_n^\dagger$, with $n \in \mathbb{N} \setminus \{0\}$ (i.e., we have an oscillator variable for each integer $n \geq 1$). These operators obey the standard commutation relations $[q, p] = i$ and $[b_n, b_{n'}^\dagger] = \delta_{nn'}$ (with all other commutators vanishing). In addition, the zero mode operator $q$ is a compact variable and is defined modulo $2\pi$, while its conjugate momentum $p$ is defined to have integer eigenvalues. This means that the Hilbert space $\mathcal{H}_{zm}$ associated with the zero mode operators $q$ and $p$ is spanned by the states $|s\rangle$, $s \in \mathbb{Z}$, which are eigenstates of $p$, $p|s\rangle = s|s\rangle$, and with $e^{\pm iq}|s\rangle = |s \pm 1\rangle$. We can also define a basis $|q\rangle$ of eigenstates of $q$, with $\langle q|s\rangle = \frac{e^{iqs}}{\sqrt{2\pi}}$, and we have $\langle q|q'\rangle = \delta_{2\pi}(q - q')$, where $\delta_{2\pi}(q - q') = \sum_{s \in \mathbb{Z}} \frac{1}{2\pi} e^{i(q-q')s}$ is the $2\pi$-periodic delta function.

In terms of these operators, the mode expansions for $\varphi(s)$ and $\vartheta(s)$ take the form

$$\varphi(s) = mq - \sum_{n=1}^\infty \frac{e^{-\frac{\epsilon n}{2}}}{\sqrt{n}} \cos(\kappa_n x)(b_n + b_n^\dagger), \tag{71a}$$

$$\vartheta(s) = (p + \delta)\frac{\pi x}{m\ell} + i \sum_{n=1}^\infty \frac{e^{-\frac{\epsilon n}{2}}}{\sqrt{n}} \sin(\kappa_n x)(b_n - b_n^\dagger),$$

where $\kappa_n = \frac{\pi n}{\ell}$. Here we have set the twist for $\vartheta$ at the two ends as a generic $\delta$; in the setup discussed in the main text, we have $\delta = 1/2$. The exponential factor $\epsilon$ is a dimensionless ultraviolet cutoff, which we will discuss in details for $m = 1$ (Appendix C) and $m \neq 1$ (Appendix F). We can see that this cutoff serves to control the oscillator sums at high momenta $\kappa_n$. Removing the cutoff corresponds to taking $a \to 0$, and one can check that in this limit the fields obey the correct commutation relations $[\vartheta(s), \partial_{s'}\varphi(s')] = [\varphi(s), \partial_{s'}\vartheta(s')] = \pi i \delta(s - s')$ (these follow from Eqs. (18) and the definition of $\varphi(s)$ and $\vartheta(s)$ in terms of $\phi_{\uparrow/\downarrow}(s)$).

Finally, as in the main text, it is convenient to define the shifted zero mode momentum operator $\tilde{p}$ via

$$\tilde{p} = p + \delta . \tag{72}$$

This will be useful because almost all of our expressions will involve the shifted momentum $\tilde{p}$ instead of the original momentum $p$. Note that, since $p$ is defined to have integer eigenvalues, the eigenvalues of $\tilde{p}$ lie in the set $\mathbb{Z} + \delta$ (the integers shifted by $\delta$).

## C Bosonization in the $m = 1$ case

In this appendix we explain how to carefully define the fermionic operators $R(s)$ and $L(s)$ in terms of the bosonic fields $\phi_{\uparrow}(s)$ and $\phi_{\downarrow}(s)$ in the non-fractional case with $m = 1$. Specifically, we define $R(s)$ and $L(s)$ as *normal-ordered exponentials* of $\phi_{\uparrow}(s)$ and $\phi_{\downarrow}(s)$, and with a dimensionful prefactor that depends on the length $\ell$. We then show that the operators $R(s)$ and $L(s)$ constructed in this way actually do obey the standard anticommutation relations of fermionic fields.

### C.1 Important identities

There are two basic identities that we will use repeatedly in the derivations in this appendix, and so we record them here for reference. Let $X$ and $Y$ be any two operators such that their commutator $[X, Y]$ is a c-number. Then we have

$$e^X e^Y = e^Y e^X e^{[X,Y]} \tag{73}$$

and

$$e^X e^Y = e^{X+Y} e^{\frac{1}{2}[X,Y]} . \tag{74}$$

### C.2 Definition of the fermion operators

We now present the definition of the operators $R(s)$ and $L(s)$. We start with the mode expansions for the fields $\phi_{\uparrow}(s)$ and $\phi_{\downarrow}(s)$, which take the form (recall that we take $m = 1$)

$$\phi_{\uparrow}(s) = q + \tilde{p}\frac{\pi x}{\ell} - \sum_{n=1}^{\infty} \frac{e^{-\frac{\epsilon n}{2}}}{\sqrt{n}} \left( e^{-i\kappa_n x} b_n + \text{h.c.} \right) , \tag{75a}$$

$$\phi_{\downarrow}(s) = q - \tilde{p}\frac{\pi x}{\ell} - \sum_{n=1}^{\infty} \frac{e^{-\frac{\epsilon n}{2}}}{\sqrt{n}} \left( e^{i\kappa_n x} b_n + \text{h.c.} \right) . \tag{75b}$$

In addition, recall that $\kappa_n = \frac{\pi n}{\ell}$ and that $\epsilon = \frac{\pi a}{\ell}$. For later use, we note here that $\phi_{\downarrow}(s) = \phi_{\uparrow}(-x)$ (this relation actually holds for any $m$ and not just $m = 1$).[4]

---

[4]It should be clear that there is no problem with plugging a negative value of the position coordinate into our mode expansions for $\phi_{\uparrow}(s)$ and $\phi_{\downarrow}(s)$.

Given these mode expansions, our definition of the fermion operators $R(s)$ and $L(s)$ is as follows. First, for any operator $\mathcal{O}$ of the form

$$\mathcal{O} = Aq + \sum_{n=1}^{\infty} B_n b_n^{\dagger} + \sum_{n=1}^{\infty} C_n b_n + Dp \;, \tag{76}$$

we define the normal-ordered exponential $: e^{\mathcal{O}} :$ by

$$: e^{\mathcal{O}} : = \; e^{Aq} e^{\sum_{n=1}^{\infty} B_n b_n^{\dagger}} e^{\sum_{n=1}^{\infty} C_n b_n} e^{Dp} \;. \tag{77}$$

Then our definition of $R(s)$ and $L(s)$ is

$$R(s) \;\; = \;\; \frac{e^{i\delta \frac{\pi x}{\ell}}}{\sqrt{2\ell}} : e^{-i\phi_{\uparrow}(s)} : \;, \tag{78a}$$

$$L(s) \;\; = \;\; \frac{e^{i\delta \frac{\pi x}{\ell}}}{\sqrt{2\ell}} : e^{i\phi_{\downarrow}(s)} : \;. \tag{78b}$$

In other words, the fermionic operators $R(s)$ and $L(s)$ are defined in terms of normal-ordered exponentials of $\phi_{\uparrow}(s)$ and $\phi_{\downarrow}(s)$, with an additional prefactor proportional to $\ell^{-\frac{1}{2}}$. This prefactor ensures that the fermionic operators have the correct units and anticommutation relations, as we show below. We can also use the definition of the normal-ordered exponential to write out these operators in more detail. For example, we find that

$$R(s) = \frac{1}{\sqrt{2\ell}} e^{-iq} \exp\left\{ i \sum_{n=1}^{\infty} \frac{e^{-\frac{\epsilon n}{2}}}{\sqrt{n}} e^{i\kappa_n x} b_n^{\dagger} \right\} \exp\left\{ i \sum_{n=1}^{\infty} \frac{e^{-\frac{\epsilon n}{2}}}{\sqrt{n}} e^{-i\kappa_n x} b_n \right\} e^{-ip\frac{\pi x}{\ell}} \;. \tag{79}$$

We now mention a few important properties of the operators $R(s)$ and $L(s)$. First, these operators, as we have defined them above, are functions of the ultraviolet cutoff $a$, although we have not indicated this dependence in our notation. Later we will show that these operators behave exactly like fermionic fields in the $a \to 0$ limit. We also note that, in our open geometry (and with our choice of boundary conditions), the fields $R(s)$ and $L(s)$ are not independent but are actually related by the identity

$$L(s) = e^{i\frac{\pi x}{\ell}} R^{\dagger}(-x) \;. \tag{80}$$

This identity can be derived by taking the Hermitian conjugate of our expression for $R(-x)$, and by using the rearrangement identity

$$e^{iq} e^{-ip\frac{\pi x}{\ell}} = e^{-ip\frac{\pi x}{\ell}} e^{iq} e^{i\frac{\pi x}{\ell}} \;, \tag{81}$$

which can be derived using Eq. (73). For our setup, however, we will focus on the interval $(0, \ell)$, in which $L$ and $R^{\dagger}$ can be treated as independent fields.

## C.3 Derivation of anticommutation relations

We now show that, in the limit $\epsilon \to 0$, the operators $R(s)$ and $L(s)$ that we defined actually do obey the standard anticommutation relations for fermionic fields. We start by deriving the anticommutator $\{R(s), R(y)\}$ between the right-moving field at two different points $x$ and $y$. For this calculation we first define four quantities (1), (2), (3), and (4) via

$$(1) \;\; = \;\; e^{-ip\frac{\pi x}{\ell}} \;, \tag{82a}$$

$$(2) \;\; = \;\; e^{-iq} \;, \tag{82b}$$

$$(3) \;\; = \;\; \exp\left\{ i \sum_{n=1}^{\infty} \frac{e^{-\frac{\epsilon n}{2}}}{\sqrt{n}} e^{-i\kappa_n x} b_n \right\} \;, \tag{82c}$$

$$(4) \;\; = \;\; \exp\left\{ i \sum_{n=1}^{\infty} \frac{e^{-\frac{\epsilon n}{2}}}{\sqrt{n}} e^{i\kappa_n y} b_n^{\dagger} \right\} \;. \tag{82d}$$

Then using Eq. (73) we find that

$$(1)(2) = (2)(1)e^{i\frac{\pi x}{\ell}} \tag{83}$$

and

$$
\begin{aligned}
(3)(4) &= (4)(3)e^{-\sum_{n=1}^{\infty} \frac{e^{-\epsilon n}}{n} e^{-i\kappa_n(x-y)}} \\
&= (4)(3)e^{\ln\left[1 - e^{-\epsilon} e^{-i\frac{\pi}{\ell}(x-y)}\right]} \\
&= (4)(3)\left[1 - e^{-\epsilon} e^{-i\frac{\pi}{\ell}(x-y)}\right] ,
\end{aligned}
\tag{84}
$$

where we used the infinite series $\ln(1-z) = -\sum_{n=1}^{\infty} z^n/n$ (valid for $|z| < 1$) to get from the first to the second line. Putting these results together yields the formula

$$R(s)R(y) = \frac{1}{2\ell} : e^{-i\phi_\uparrow(s) - i\phi_\uparrow(y)} : \left[e^{i\frac{\pi x}{\ell}} - e^{-\epsilon} e^{i\frac{\pi y}{\ell}}\right] . \tag{85}$$

By examining the term in square brackets, which tends to $e^{i\frac{\pi x}{\ell}} - e^{i\frac{\pi y}{\ell}}$ as $a \to 0$, we can see that

$$\lim_{a \to 0} \{R(s), R(y)\} = 0 \ \ \forall \ \ x, \ y , \tag{86}$$

which is the expected anticommutator for a fermionic field with itself.

Next, we consider the anticommutator of $R(s)$ with $R^\dagger(y)$. For this calculation we define the operator $A(s)$ by

$$A(s) = \sum_{n=1}^{\infty} \frac{e^{-\frac{\epsilon n}{2}}}{\sqrt{n}} e^{i\kappa_n x} b_n^\dagger , \tag{87}$$

and so

$$A^\dagger(s) = \sum_{n=1}^{\infty} \frac{e^{-\frac{\epsilon n}{2}}}{\sqrt{n}} e^{-i\kappa_n x} b_n . \tag{88}$$

In terms of this operator we can rewrite $R(s)$ and $R^\dagger(y)$ as

$$
\begin{aligned}
R(s) &= \frac{1}{\sqrt{2\ell}} e^{-iq} e^{iA(s)} e^{iA^\dagger(s)} e^{-ip\frac{\pi x}{\ell}} , \tag{89} \\
R^\dagger(y) &= \frac{1}{\sqrt{2\ell}} e^{ip\frac{\pi y}{\ell}} e^{-iA(y)} e^{-iA^\dagger(y)} e^{-iq} . \tag{90}
\end{aligned}
$$

Then, using similar rearrangement identities as in our previous calculation (using Eq. (73) again), we obtain the formulas

$$R(s)R^\dagger(y) = \frac{1}{2\ell} e^{-ip\frac{\pi}{\ell}(x-y)} e^{iA(s)-iA(y)} e^{iA^\dagger(s)-iA^\dagger(y)} \frac{e^{-i\frac{\pi}{\ell}(x-y)}}{1 - e^{-\epsilon} e^{-i\frac{\pi}{\ell}(x-y)}} \tag{91}$$

and

$$R^\dagger(y)R(s) = \frac{1}{2\ell} e^{-ip\frac{\pi}{\ell}(x-y)} e^{iA(s)-iA(y)} e^{iA^\dagger(s)-iA^\dagger(y)} \frac{1}{1 - e^{-\epsilon} e^{i\frac{\pi}{\ell}(x-y)}} . \tag{92}$$

For the anticommutator we then find the formula

$$\{R(s), R^\dagger(y)\} = e^{-ip\frac{\pi}{\ell}(x-y)} e^{iA(s)-iA(y)} e^{iA^\dagger(s)-iA^\dagger(y)} d(x-y; \epsilon) , \tag{93}$$

where we defined the function $d(x - y; a)$ by

$$d(x-y; \epsilon) = \frac{1}{2\ell}\left[\frac{e^{-i\frac{\pi}{\ell}(x-y)}}{1 - e^{-\epsilon} e^{-i\frac{\pi}{\ell}(x-y)}} + \frac{1}{1 - e^{-\epsilon} e^{i\frac{\pi}{\ell}(x-y)}}\right] . \tag{94}$$

We now examine the properties of the function $d(x - y; a)$. For $a > 0$ we can expand the denominators in $d(x - y; a)$ as geometric series to obtain

$$d(x - y; \epsilon) = \frac{1}{2\ell} \left[ e^{-i\frac{\pi}{\ell}(x-y)} \sum_{n=0}^{\infty} e^{-\epsilon n} e^{-i\frac{\pi n}{\ell}(x-y)} + \sum_{n=0}^{\infty} e^{-\epsilon n} e^{i\frac{\pi n}{\ell}(x-y)} \right].$$

From this expression we can see that

$$\lim_{\epsilon \to 0} d(x - y; \epsilon) = \delta_{2\ell}(x - y), \tag{95}$$

where

$$\delta_{2\ell}(x - y) = \frac{1}{2\ell} \sum_{n \in \mathbb{Z}} e^{i\frac{2\pi n}{2\ell}(x-y)} \tag{96}$$

is the $2\ell$-periodic delta function. Since this function is zero for $x \neq y$ modulo $2\ell$, and since all of the prefactors from Eq. (93) are equal to 1 when $x = y$ modulo $2\ell$, we then find that

$$\lim_{a \to 0} \{R(s), R^{\dagger}(y)\} = \delta_{2\ell}(x - y). \tag{97}$$

Therefore we have proven that, in the limit $\epsilon \to 0$, the operator $R(s)$ obeys the standard anticommutation relations for a fermionic field operator. Since the anti-commutation relation (97) holds down to the smallest length scales, one can identify the ultraviolet cutoff $\epsilon$ in the bosonic theory as that in the fermionic theory, which is given by the lattice constant $a$ as

$$\epsilon = \frac{\pi a}{\ell}. \tag{98}$$

For the left-moving field $L(s)$, we can use our results for $R(s)$ and the relation (80) between $L(s)$ and $R(s)$ to immediately conclude that $L(s)$ also obeys the standard anticommutation relations for a fermionic field operator.

Finally, there is one more interesting anticommutation relation that we can obtain for our system with open boundary conditions. Since in this case the left- and right-moving fermionic fields are not independent, we find that, for $x, y \in (0, \ell)$,

$$\begin{aligned} \{R(s), L(y)\} &= e^{i\frac{\pi y}{\ell}} \{R(s), R^{\dagger}(-y)\} \\ &= e^{i\frac{\pi y}{\ell}} \delta_{2\ell}(x + y) \\ &= 0, \end{aligned} \tag{99}$$

where the last line holds since $x + y \neq 0$ for $x, y \in (0, \ell)$. For $x, y \in (0, \ell)$ we also have

$$\begin{aligned} \{R(s), L^{\dagger}(y)\} &= e^{-i\frac{\pi y}{\ell}} \{R(s), R(-y)\} \\ &= 0. \end{aligned} \tag{100}$$

Therefore, for our system with open boundary conditions, we find that the left- and right-moving fermionic fields already have the correct anticommutation relations, and we do not need to include any extra *Klein factors* to ensure that $R(s)$ and $L(y)$ (and $R(s)$ and $L^{\dagger}(y)$) anticommute.

## C.4 Correlation functions in the free theory

To complete this section we present the formulas for the two-point correlation functions of $R(s)$ and $L(s)$ in the free theory before adding any perturbation terms. The form of these correlation functions will complete the demonstration that we have correctly constructed the fermionic operators from the bosonic fields.

We consider the free bosonic vacuum $|0\rangle$ that satisfies $p|0\rangle = 0$ and $b_n|0\rangle = 0$ for all $n \in \{1, 2, 3, \dots\}$. Using our previous rearrangement of the product $R^\dagger(y)R(s)$, we find that in this state we have

$$\langle 0|R^\dagger(y)R(s)|0\rangle = \frac{1}{2\ell} \frac{1}{\left[1 - e^{-\epsilon}e^{i\frac{\pi}{\ell}(x-y)}\right]} , \tag{101}$$

and by taking the complex conjugate we find that

$$\langle 0|R^\dagger(s)R(y)|0\rangle = \frac{1}{2\ell} \frac{1}{\left[1 - e^{-\epsilon}e^{-i\frac{\pi}{\ell}(x-y)}\right]} . \tag{102}$$

To check that this formula makes sense, we can investigate its behavior in the bulk of the system, which corresponds to taking the limit $\ell \to \infty$ while keeping $x - y$, $\delta$. We also hold the ultraviolet cutoff $a$ fixed in this limit (although it is safe to take it to zero at this point). Then in this limit we find that

$$\langle 0|R^\dagger(s)R(y)|0\rangle \to \frac{1}{2\pi} \frac{1}{\left[i(x-y) + a\right]} , \tag{103}$$

which is the correct bulk correlation function of a free right-moving fermion in one spatial dimension (with the correct normalization).

We can now do a similar calculation for the left-moving fermion. Using the relation between $R(s)$ and $L(s)$, we first find that

$$\langle 0|L^\dagger(s)L(y)|0\rangle = e^{-i\frac{\pi}{\ell}(x-y)}\langle 0|R(-x)R^\dagger(-y)|0\rangle . \tag{104}$$

Then, using our previous rearrangement of $\langle 0|R(s)R^\dagger(y)|0\rangle$, we find that

$$\langle 0|L^\dagger(s)L(y)|0\rangle = \frac{1}{2\ell} \frac{1}{\left[1 - e^{-\epsilon}e^{i\frac{\pi}{\ell}(x-y)}\right]} . \tag{105}$$

If we now take the bulk limit then in this case we find that

$$\langle 0|L^\dagger(s)L(y)|0\rangle \to \frac{1}{2\pi} \frac{1}{\left[-i(x-y) + a\right]} , \tag{106}$$

which is the correct bulk correlation function of a free left-moving fermion in one spatial dimension.

# D  The variational approximation

In this appendix we explain the variational approximation that we use to study the ground states of the domain wall Hamiltonian from Eq. (33a) of Sec. 3.2 of the main text. This variational approximation leads us to an effective Hamiltonian that only involves the zero mode operators $q$ and $p$ from the mode expansions of $\varphi(s)$ and $\vartheta(s)$, and in the later appendices we analyze this effective Hamiltonian in detail and use it to make predictions for the physical properties of the original domain wall model.

The variational method is familiar from quantum mechanics. It allows one to obtain information about the ground state of a system by making sufficiently clever guesses for the form of the ground state wave function. Here we apply this method to study the ground state of a quantum field theory, and in this setting there are additional complications associated with divergences present in a quantum field theory without a proper cutoff. Therefore, we perform our variational calculation for the domain wall model with a finite ultraviolet cutoff $a$. Then, at the end of the calculation, we consider the system with a small but finite value of $a$, as

in condensed matter systems it is sensible to keep a finite ultraviolet cutoff $a$ (which can be intuitively thought of as being related to the scale of the crystal lattice).

Our starting point is the full Hamiltonian for the domain wall model. We denote this Hamiltonian by $H(a)$ to indicate that we are working with a finite ultraviolet cutoff $a > 0$. As in Sec. 2, this Hamiltonian takes the form

$$H(a) = H_0(a) - B \int_0^\ell ds \; \left( R^\dagger(s) L(s) + \text{h.c.} \right) . \tag{107}$$

Here, $R(s)$ and $L(s)$ are the fermionic operators from Eq. (78) that we constructed from the bosonic fields $\phi_\uparrow(s)$ and $\phi_\downarrow(s)$. For simplicity we also assume that the magnetic field points along the positive $x$-axis, so that $\tau = 0$ in our previous notation. Finally, $H_0(a)$ is the part of the Hamiltonian that contains the kinetic energy term and the density-density interactions.

To prepare for our variational approximation, we first write out the term $H_0(a)$ using our mode expansions for $\varphi(s)$ and $\vartheta(s)$. We find that

$$
\begin{aligned}
H_0(a) = {} & \frac{v K \pi}{2\ell} \tilde{p}^2 \\
& + \frac{v}{4} \left( \frac{1}{K} + K \right) \sum_{n=1}^\infty e^{-\epsilon n} \kappa_n (b_n^\dagger b_n + b_n b_n^\dagger) \\
& + \frac{v}{4} \left( \frac{1}{K} - K \right) \sum_{n=1}^\infty e^{-\epsilon n} \kappa_n (b_n b_n + b_n^\dagger b_n^\dagger) .
\end{aligned}
\tag{108}
$$

The oscillator part of $H_0(a)$ can be diagonalized by making a Bogoliubov transformation to new oscillator variables $a_n$ defined by

$$a_n = \cosh(\eta) b_n + \sinh(\eta) b_n^\dagger , \tag{109}$$

where the real parameter $\eta$ is related to $K$ as

$$e^{-2\eta} = K. \tag{110}$$

In terms of these new variables, we find that

$$H_0(a) = \frac{v K \pi}{2\ell} \tilde{p}^2 + \frac{v}{2} \sum_{n=1}^\infty e^{-\epsilon n} \kappa_n (a_n^\dagger a_n + a_n a_n^\dagger) .$$

For later use we also note the reverse Bogoliubov transformation,

$$b_n = \cosh(\eta) a_n - \sinh(\eta) a_n^\dagger , \tag{111}$$

which allows us to express $b_n$ in terms of $a_n$ and $a_n^\dagger$.

As we mentioned in the main text, the Zeeman term $\sim B \int R^\dagger L$ gaps out the region and makes the oscillator modes massive. To find a suitable trial state for the oscillator part of the Hilbert space, an additional Bogoliubov transformation is needed. To this end we introduce yet another set of oscillator variables, which we denote by $\tilde{a}_n$ (with Hermitian conjugates $\tilde{a}_n^\dagger$). These will be related to the $a_n$ oscillators via the Bogoliubov transformation

$$a_n = \cosh(\zeta_n) \tilde{a}_n - \sinh(\zeta_n) \tilde{a}_n^\dagger , \tag{112}$$

where we have allowed the parameter $\zeta_n \in \mathbb{R}$ that determines the transformation to depend on the index $n$. Using these new variables, we can rewrite $H_0(a)$ in the form

$$
\begin{aligned}
H_0(a) =\ & \frac{vK\pi}{2\ell}\tilde{p}^2 \\
& + \frac{v}{2}\sum_{n=1}^{\infty} e^{-\epsilon n}\kappa_n \cosh(2\zeta_n)(\tilde{a}_n^{\dagger}\tilde{a}_n + \tilde{a}_n\tilde{a}_n^{\dagger}) \\
& - \frac{v}{2}\sum_{n=1}^{\infty} e^{-\epsilon n}\kappa_n \sinh(2\zeta_n)(\tilde{a}_n\tilde{a}_n + \tilde{a}_n^{\dagger}\tilde{a}_n^{\dagger})\ .
\end{aligned}
\tag{113}
$$

We also find that $b_n$ is related to $\tilde{a}_n$ via the relation

$$
b_n = \cosh(\eta + \zeta_n)\tilde{a}_n - \sinh(\eta + \zeta_n)\tilde{a}_n^{\dagger}\ ,
\tag{114}
$$

which follows from identities for the hyperbolic trigonometric functions.

We now discuss our choice of variational trial state. Let $|0,\zeta\rangle$ be the Fock vacuum state annihilated by all the $\tilde{a}_n$,

$$
\tilde{a}_n|0,\zeta\rangle = 0\ \forall\ n\ .
\tag{115}
$$

Note also that, as we are now working in terms of the $\tilde{a}_n$ oscillator variables, the full Hilbert space $\mathcal{H}_{\text{tot}}$ of our domain wall model is equal to the tensor product

$$
\mathcal{H}_{\text{tot}} = \mathcal{H}_{\text{zm}} \otimes \mathcal{H}_{\text{F}}\ ,
\tag{116}
$$

where $\mathcal{H}_{\text{F}}$ is the Fock space generated by the action of the raising operators $\tilde{a}_n^{\dagger}$ on the Fock vacuum $|0,\zeta\rangle$, and $\mathcal{H}_{\text{zm}}$ is the Hilbert space for the zero modes ($q$ and $p$ act within $\mathcal{H}_{\text{zm}}$). The trial ground state $|\Psi\rangle$ that we consider respects the tensor product structure of the Hilbert space and it takes the tensor product form

$$
|\Psi\rangle = |\psi\rangle \otimes |0,\zeta\rangle\ ,
\tag{117}
$$

where $|\psi\rangle$ is a state in the zero mode Hilbert space $\mathcal{H}_{\text{zm}}$ and $|0,\zeta\rangle$ is the Fock vacuum for the $\tilde{a}_n$ variables.

The nontrivial part of our variational calculation is the problem of finding the parameters $\zeta_n$ and the zero mode state $|\psi\rangle \in \mathcal{H}_{\text{zm}}$ that minimize the energy expectation value $\langle\Psi|H(a)|\Psi\rangle$. In fact, we will not carry out this optimization procedure completely on the $\zeta_n$ parameters. Instead, we will use a heuristic argument to obtain the behavior of the energy expectation value with the correct choice of $\zeta_n$.

To proceed with the variational calculation we need to compute the expectation value $\langle\Psi|H(a)|\Psi\rangle$ and then consider this expectation value in the small $a$ limit. We now present this calculation, omitting many of the details since the required manipulations are similar to the ones we used in Appendix C to prove the bosonization formulas. For the kinetic term we find that

$$
\langle\Psi|H_0(a)|\Psi\rangle = \frac{vK\pi}{2\ell}\langle\psi|\tilde{p}^2|\psi\rangle + \frac{v}{2}\sum_{n=1}^{\infty} e^{-\epsilon n}\kappa_n \cosh(2\zeta_n)\ ,
\tag{118}
$$

where the second term here is the vacuum energy for the $\tilde{a}_n$ oscillators. For the Zeeman term we find that

$$
\langle\Psi|R^{\dagger}(s)L(s)|\Psi\rangle = \frac{1}{2}f(x;a;\zeta)\langle\psi|e^{i2q}|\psi\rangle\ ,
\tag{119}
$$

where the function $f(x;a;\zeta)$ is given by

$$
f(x;a;\zeta) = \frac{1}{\ell}\frac{1}{1-e^{-\epsilon}}\left[\frac{1-e^{-\epsilon-i\frac{2\pi x}{\ell}}}{1-e^{-\epsilon+i\frac{2\pi x}{\ell}}}\right]^{\frac{1}{2}} e^{i\frac{2\pi x}{\ell}} e^{-\sum_{n=1}^{\infty}\frac{e^{-\epsilon n}}{n}e^{-2(\eta+\zeta_n)}(1+\cos(2\kappa_n x))}\ .
\tag{120}
$$

We note that the summation in the exponent of the last factor

$$\sum_{n=1}^{\infty} \frac{e^{-\epsilon n}}{n} e^{-2(\eta+\zeta_n)} \tag{121}$$

is logarithmical, with the series effectively truncated by the factors $e^{-\epsilon n}$ and $e^{-2\zeta_n}$. The first factor $e^{-\epsilon n}$, with $\epsilon = \pi a/\ell$, is a ultraviolet cutoff for the mode number, $n \lesssim \ell/a$. The second factor $e^{-2\zeta_n}$, on the other hand, is due to the quasiparticle mass gap $\Delta_Z$ induced by the Zeeman field $B$. Obviously we have

$$\Delta_Z = B \text{ for free fermions,} \tag{122}$$

but in general $\Delta_Z$ is renormalized by interactions and is a parameter determined by the choice of $\{\zeta_n\}$. The $e^{-2\zeta_n}$ factor effectively serves as an infrared cutoff for the mode number $n$, as for high enough modes $\kappa_n \gg \Delta_Z$ the effect of the Zeeman field can be neglected. This means that the correct choice of $\zeta_n$ sets a lower limit of summation, $n \gtrsim \Delta_Z \ell$. Therefore in the regime

$$a \ll 1/\Delta_Z \ll \ell, \tag{123}$$

the following summation can be approximated by

$$\sum_{n=1}^{\infty} \frac{e^{-\epsilon n}}{n} e^{-2(\eta+\zeta_n)} \approx -K \ln(\Delta_Z a), \tag{124}$$

where we have used Eq. (110). Plugging (124) into (120) we obtain

$$f(x;a;\zeta) = \Delta_Z^K a^{K-1} f_0(x;a;\zeta), \tag{125}$$

where $f_0$ is a well-behaved $\mathcal{O}(1)$ function. With this choice, we then define an energy scale $\beta$ via

$$\beta = B \int_0^{\ell} ds \, f(x;a;\zeta), \tag{126}$$

and we find that at small $a$ we have an extensive behavior

$$\beta \sim B \Delta_Z^K a^{K-1} \ell. \tag{127}$$

For a free fermion system with $K=1$, this energy is $\sim B^2 \ell$. Using these results, we can now complete our calculation of $\langle \Psi | H(a) | \Psi \rangle$. We find that

$$\langle \Psi | H(a) | \Psi \rangle = \alpha \langle \psi | \tilde{p}^2 | \psi \rangle - \beta \langle \psi | \cos(2q) | \psi \rangle, \tag{128}$$

where the coefficients $\alpha$ and $\beta$ are given by

$$\alpha = \frac{\nu K \pi}{2\ell}, \tag{129a}$$

$$\beta \sim B \Delta_Z^K a^{K-1} \ell, \tag{129b}$$

and we have omitted the $c$-number vacuum energy term for the $\tilde{a}_n$ oscillators. This result tells us that for our variational approximation the zero mode state $|\psi\rangle$ should be chosen to be the lowest energy state of the effective zero mode Hamiltonian

$$H_{\text{eff}} = \alpha \tilde{p}^2 - \beta \cos(2q). \tag{130}$$

We notice that in the limit $\beta \gg \alpha$, this Hamiltonian describes an approximate harmonic oscillator, with energy level spacing given by $\sim \sqrt{\alpha\beta}$. These energy levels form a Fock space of zero modes, now with mass $\sim \sqrt{\alpha\beta}$. We can identify the mass scale of the former zero modes with that of the oscillator modes:

$$\Delta_Z \sim \sqrt{\alpha\beta}. \tag{131}$$

Self-consistency between Eqs. (129) and (131) fixes the scale of $\Delta_Z$ as

$$\Delta_Z \sim B\,(Ba)^{\frac{K-1}{2-K}}. \tag{132}$$

Notably, we indeed recover $\Delta_Z = B$ for the free fermion case $K = 1$. With Eq. (132), the condition (123) translates to

$$\frac{1}{a}\left(\frac{a}{\ell}\right)^{2-K} \ll B \ll \frac{1}{a}, \tag{133}$$

which requires $K < 2$. For $K \geq 2$ a separate variational ansatz is needed, which we postpone to future studies. As a sanity check, we see that the condition $\beta \gg \alpha$ we needed is precisely one of the conditions in Eq. (133). We note that the condition $K < 2$ is consistent with Kosterlitz renormalization group results on the sine-Gordon model in infinite spacetime, under which the cosine term is a relevant perturbation.

We can rewrite the parameters $\alpha$ and $\beta$ as

$$\alpha = \frac{vK\pi}{2\ell}, \tag{134a}$$

$$\beta \sim B^{\frac{2}{2-K}} a^{\frac{2K-2}{2-K}} \ell. \tag{134b}$$

As we discuss in the next appendix, $H_{\text{eff}}$ is closely related to Mathieu's equation, and so we can use known results on that equation to study $H_{\text{eff}}$ and solve our variational problem. In that appendix we present error estimates for various quantities, and those error estimates are *exponentially* small in $\ell$. The key to obtaining that scaling for the error estimates is the fact that the parameters $\alpha$ and $\beta$ in $H_{\text{eff}}$ satisfy the relation

$$\lambda^2 \equiv \frac{\beta}{\alpha} \sim B^{\frac{2}{2-K}} a^{\frac{2K-2}{2-K}} \ell^2. \tag{135}$$

It is convenient to define a correlation length such that $\lambda \propto \ell/\xi$, and we have

$$\xi/a \sim \left(\frac{1}{Ba}\right)^{\frac{1}{2-K}}, \tag{136}$$

which diverges at $K \to 2$ obeying the familiar Kosterlitz-Thouless scaling behavior.

The full Hamiltonian $H(a)$ and the effective Hamiltonian $H_{\text{eff}}$ both have a $\mathbb{Z}_2$ symmetry generated by the operator $e^{i\pi p}$ (the symmetry is $\mathbb{Z}_2$ because $p$ has integer eigenvalues). This means that the Hilbert space of the domain wall model is broken up into two different sectors, where the states in each sector have opposite eigenvalues ($\pm 1$) of $e^{i\pi p}$. It also means that we should carry out our variational calculation separately in each sector to study the ground state of the Hamiltonian within each sector.

We can gain a more physical understanding of this $\mathbb{Z}_2$ symmetry by noting that $e^{i\pi p}$ is proportional to the parity $e^{i\pi S}$ of the total spin $S$ in the FM region, since $S$ is given explicitly by

$$\begin{aligned} S &= \frac{1}{2\pi}\int_0^\ell ds\,\left[\partial_s\phi_\uparrow(s) - \partial_s\phi_\downarrow(s)\right] \\ &= p + \delta \\ &= \tilde{p}. \end{aligned} \tag{137}$$

From a physical point of view this makes sense since the spin parity $e^{i\pi S}$ commutes with the Hamiltonian in the FM region. Therefore in what follows it is convenient for us to label the two sectors of the Hilbert space by a "lattice momentum" $k \in [-1, 1) \cap (\mathbb{Z} + \delta)$, such that any given state is an eigenstate of $e^{i\pi\tilde{p}}$ with eigenvalue $e^{i\pi k}$.[5] Note that, since $p$ has integer eigenvalues, there are only two such values of $k$ in the set $[-1, 1) \cap (\mathbb{Z} + \delta)$. For example, in the case of $\delta = \frac{1}{2}$, which is our main interest, we have $k \in \{-\frac{1}{2}, \frac{1}{2}\}$.

Our variational method can be used to study the lowest energy states of $H(a)$ with all possible eigenvalues of $e^{i\pi\tilde{p}}$ (i.e., all possible values of the lattice momentum $k$). In particular, we are interested in estimating the energy splitting between the lowest energy states of $H(a)$ with different $e^{i\pi\tilde{p}}$ eigenvalues. We therefore define separate variational trial states $|\Psi_k\rangle = |\psi_k\rangle \otimes |0, \zeta\rangle$ for each allowed value of $k$, where $|\psi_k\rangle$ should be chosen to be the ground state of $H_{\text{eff}}$ in the sector of $\mathcal{H}_{\text{zm}}$ with $e^{i\pi\tilde{p}} = e^{i\pi k}$.

Let $E_k$ be the energy of the ground state $|\psi_k\rangle$ of $H_{\text{eff}}$ in the sector with $e^{i\pi\tilde{p}} = e^{i\pi k}$,

$$H_{\text{eff}}|\psi_k\rangle = E_k|\psi_k\rangle \quad , \quad e^{i\pi\tilde{p}}|\psi_k\rangle = e^{i\pi k}|\psi_k\rangle , \tag{138}$$

and let $k_1$ and $k_2$ be the two allowed values of $k$ in the set $[-1, 1) \cap (\mathbb{Z} + \delta)$. Our variational estimate for the energy splitting $\Delta E$ between the ground states of the domain wall model in the sectors with $e^{i\pi\tilde{p}} = e^{i\pi k_1}$ and $e^{i\pi\tilde{p}} = e^{i\pi k_2}$ is then given by

$$\Delta E \approx \left| \left( \langle \Psi_{k_1}|H(a)|\Psi_{k_1}\rangle - \langle \Psi_{k_2}|H(a)|\Psi_{k_2}\rangle \right) \right|$$
$$= |E_{k_1} - E_{k_2}| . \tag{139}$$

In the next appendix we use known results on Mathieu's differential equation to show that $|E_{k_1} - E_{k_2}|$ is exponentially small in $\ell$. Then our variational approximation predicts that the ground states of the domain wall model with different $e^{i\pi\tilde{p}}$ eigenvalues are very nearly degenerate for large $\ell$. In addition, for the special case where $\delta = \frac{1}{2}$, which is our main interest in this paper, we show in Appendices G and H that the full domain wall Hamiltonian $H(a)$ has an additional discrete symmetry that guarantees that every eigenstate of $H(a)$ has a partner with the exact same energy, and so the ground state of of $H(a)$ is exactly degenerate for any $\ell$ (not just approximately degenerate for large $\ell$).

# E  Results from the variational approximation (Mathieu's equation)

In this appendix we present our main results on the domain wall model that we described in Sec. 3.2. We first apply known mathematical results on Mathieu's equation to understand the ground states of the effective zero mode Hamiltonian $H_{\text{eff}}$. We then use these results in our variational approximation to obtain nontrivial predictions for certain properties of the domain wall model, including the finite-size splitting of the nearly degenerate ground states (when $\delta \neq 1/2$ – there is an exact degeneracy at $\delta = \frac{1}{2}$ that we discuss in Appendices G and H), the correlation functions of the fermionic operators, and the Berry phase for certain adiabatic processes involving the external magnetic field.

## E.1  Bound on $|E_{k_1} - E_{k_2}|$ for $H_{\text{eff}}$ and estimate of the splitting $\Delta E$

We first explain how known results on Mathieu's differential equation can be used to bound the difference $|E_{k_1} - E_{k_2}|$ between the energies of the lowest energy states of $H_{\text{eff}}$ in the two sectors with different $e^{i\pi\tilde{p}}$ eigenvalue. We start by explaining the relation between $H_{\text{eff}}$ and Mathieu's

---

[5]Note that $k$ should lie in $[-1, 1)$ because this is the first Brillouin zone for a lattice with lattice spacing equal to $\pi$.

differential equation. We first note that, by construction, $p$ has integer eigenvalues. It follows from this that all states in the zero mode Hilbert space $\mathcal{H}_{zm}$ are invariant under the action of $e^{i2\pi p}$, which is the operator that translates $q$ by $2\pi$, $e^{i2\pi p} q e^{-i2\pi p} = q + 2\pi$. Therefore, if $|\psi\rangle$ is any state in $\mathcal{H}_{zm}$, then its wave function $\psi(q) = \langle q|\psi\rangle$ is $2\pi$-periodic in $q$, $\psi(q + 2\pi) = \psi(q)$. Next, as we discussed in the previous appendix, $H_{\text{eff}}$ also commutes with the operator $e^{i\pi p}$ that translates $q$ by $\pi$, $e^{i\pi p} q e^{-i\pi p} = q + \pi$. Accordingly, all eigenstates of $H_{\text{eff}}$ can be chosen to be eigenstates of $e^{i\pi p}$, as we have discussed (and we actually labeled states by their eigenvalue of the closely related operator $e^{i\pi\tilde{p}}$).

Let $|\psi\rangle$ be an eigenstate of $H_{\text{eff}}$ with energy $E$. Then the wave function $\psi(q)$ satisfies the Schrodinger equation

$$\alpha\left(-i\frac{d}{dq} + \delta\right)^2 \psi(q) - \beta\cos(2q)\psi(q) = E\psi(q) , \tag{140}$$

where $p$ has become the differential operator $-i\frac{d}{dq}$. If we define a new wave function $\chi(q)$ by

$$\psi(q) = e^{-i\delta q}\chi(q) , \tag{141}$$

then we find that $\chi(q)$ satisfies

$$-\alpha\chi''(q) - \beta\cos(2q)\chi(q) = E\chi(q) , \tag{142}$$

where $\chi'(q) = \frac{d\chi(q)}{dq}$. This equation can be brought into the standard form of Mathieu's equation by dividing through by $\alpha \neq 0$ to obtain

$$-\chi''(q) - \lambda^2\cos(2q)\chi(q) = \mathcal{E}\chi(q) , \tag{143}$$

where we remind $\lambda^2 = \beta/\alpha$ and $\mathcal{E} = E/\alpha$. In addition, the $2\pi$-periodicity of $\psi(q)$ implies that $\chi(q)$ obeys the periodicity condition

$$\chi(q + 2\pi) = e^{i2\pi\delta}\chi(q) . \tag{144}$$

To apply known results from the study of the Mathieu's equation, we need to study the behavior of $\chi(q)$ under translations by $\pi$, which is the period of the potential $\cos(2q)$ that appears in the equation. This behavior will depend on the eigenvalue of a given state under the action of the operator $e^{i\pi p}$. In particular, for a state $|\psi_k\rangle$ that satisfies $e^{i\pi p}|\psi_k\rangle = e^{i\pi(k-\delta)}|\psi_k\rangle$ (so that $e^{i\pi\tilde{p}}|\psi_k\rangle = e^{i\pi k}|\psi_k\rangle$), we find that the corresponding function $\chi_k(q) = e^{i\delta q}\psi_k(q) = e^{i\delta q}\langle q|\psi_k\rangle$ satisfies the periodicity condition

$$\begin{aligned}
\chi_k(q + \pi) &= e^{i\delta(q+\pi)}\psi_k(q + \pi) \\
&= e^{i\pi k}\chi_k(q) , \tag{145}
\end{aligned}$$

and this simple relation explains why we chose to label our states by their eigenvalue of $e^{i\pi\tilde{p}}$ instead of their eigenvalue of $e^{i\pi p}$.

It is known from Floquet theory (similar to Bloch's theorem from condensed matter physics), that the spectrum of the Mathieu operator $-\frac{d^2}{dq^2} - \lambda^2\cos(2q)$ is divided into distinct energy bands. In addition, the eigenfunctions within each energy band are labeled by a wave number $k \in [-1, 1)$, which corresponds to the Brillouin zone of a one-dimensional lattice with period $\pi$. An eigenfunction $\chi_k(q)$ characterized by the wave number $k$ obeys exactly the periodicity condition from Eq. (145). From this we see that the lowest energy state of $H_{\text{eff}}$ in the sector with $e^{i\pi\tilde{p}} = e^{i\pi k}$ corresponds exactly to the eigenfunction labeled by $k$ within the lowest band of the spectrum of $-\frac{d^2}{dq^2} - \lambda^2\cos(2q)$. Therefore, the energy splitting $|E_{k_1} - E_{k_2}|$ between the two lowest energy states of $H_{\text{eff}}$ with different $e^{i\pi\tilde{p}}$ eigenvalues is certainly less than $\alpha$ times

the width $|W_0(\lambda)|$ of the lowest band of the Mathieu operator $-\frac{d^2}{dq^2} - \lambda^2 \cos(2q)$ (we multiply by $\alpha$ because $E = \alpha \mathcal{E}$).

An asymptotic formula for the width $|W_0(\lambda)|$ at large $\lambda$ was obtained in Ref. [84] (see also Ref. [85] for a convenient summary of the properties of the spectrum of the Mathieu operator). It takes the form[6]

$$|W_0(\lambda)| = \frac{2^{\frac{19}{4}}}{\pi^{\frac{1}{2}}} \lambda^{\frac{3}{2}} e^{-\lambda\sqrt{8}} \left[ 1 + O(\lambda^{-\frac{1}{2}}) \right] . \tag{146}$$

The key feature of this formula is the factor of $e^{-\lambda\sqrt{8}}$. The presence of this factor implies that, when $\lambda$ is large, the width $|W_0(\lambda)|$ of the lowest band is exponentially small in $\lambda$. Now for our model (which we obtained from our variational approximation), this means that the splitting $|E_{k_1} - E_{k_2}|$ is exponentially small in $\ell$,

$$|E_{k_1} - E_{k_2}| \lesssim \text{constant} \times e^{-\frac{\ell}{\xi}} , \tag{147}$$

where $\xi \equiv \lambda\sqrt{8}$ is the correlation length given by Eq. (136)). Thus, our variational approximation predicts that for large $\ell$ the energy splitting $\Delta E$ of the two ground states in our domain wall model is exponentially small in the length $\ell$ of the FM region. For the free fermion case with $K = 1$, the correlation length is given by $\xi \sim 1/B$, consistent with the decaying behavior from solving the Dirac equation with a mass domain wall.

Finally, we close this section by noting that the case we are most interested in in this paper is the special case where $\delta = \frac{1}{2}$. In Appendices G and H we will show that in this case the two ground states of our domain wall model are *exactly* degenerate, and not just approximately degenerate as we have predicted here for a general $\delta$.

## E.2 Approximate form of $\chi_k(q)$ at large $\lambda$

For the Berry phase calculation later in this appendix we will need to understand the form of the eigenfunctions $\chi_k(q)$ of the Mathieu operator in the limit of large $\lambda$ (we referred to this as the "tight-binding" limit in the main text). Therefore, in this subsection we review some known facts about $\chi_k(q)$ in this limit.

When $\lambda$ is large, the eigenfunctions of the Mathieu operator in its lowest band are well-approximated by a weighted sum of Gaussians localized in each valley of the $\cos(2q)$ potential (see the proof of Theorem 1 in Ref. [86]). These approximate eigenfunctions can be constructed as follows. We first expand $\cos(2q)$ to order $q^2$ about its minimum at $q = 0$ and study the resulting approximate Mathieu operator near $q = 0$. Up to a constant, we find the operator $-\frac{d^2}{dq^2} + 2\lambda^2 q^2$, and it is well-known that the lowest energy eigenfunction of this operator is a Gaussian of the form

$$\chi_0(q) = \left( \frac{\lambda\sqrt{2}}{\pi} \right)^{\frac{1}{4}} e^{-\frac{\lambda}{\sqrt{2}} q^2} , \tag{148}$$

where we have chosen the coefficient so that $\int_{-\infty}^{\infty} dq \, |\chi_0(q)|^2 = 1$.

Let $\chi_k(q)$ be the eigenfunction in the lowest band of the Mathieu operator and obeying the periodicity condition $\chi_k(q + \pi) = e^{ik\pi} \chi_k(q)$. At large $\lambda$, this eigenfunction is given approximately by the periodic sum

$$\chi_k(q) = \frac{1}{\sqrt{2}} \sum_{n \in \mathbb{Z}} e^{ikn\pi} \chi_0(q - n\pi) , \tag{149}$$

---

[6]In Ref. [84] the bandwidth $|W_0(\lambda)|$ was denoted by $|B_0(\lambda)|$, but we use $|W_0(\lambda)|$ here to avoid confusion with the magnetic field in our problem.

which contains all translations of $\chi_0(q)$ by integer multiples of $\pi$, with the translation by $n\pi$ accompanied by the $k$-dependent phase factor $e^{ikn\pi}$. The factor of $\sqrt{2}$ is included here so that $\chi_k(q)$ obeys the normalization condition[7]

$$\int_0^{2\pi} dq \, \overline{\chi_k(q)} \chi_k(q) = 1 + O(e^{-\frac{\pi^2}{2\sqrt{2}}\lambda}) \,, \tag{150}$$

where the integral is restricted to $[0, 2\pi)$ because this is the physical range of $q$ in our problem. To understand the error estimate here, note that the overlap of $\chi_0(q)$ and $\chi_0(q-d)$ is exponentially small in $\lambda$,

$$\int_{-\infty}^{\infty} dq \, \chi_0(q) \chi_0(q-d) = e^{-\frac{d^2}{2\sqrt{2}}\lambda} \,. \tag{151}$$

This means that the dominant contribution to $\int_0^{2\pi} dq \, \overline{\chi_k(q)} \chi_k(q)$ comes from the overlap between Gaussians in the same position, while the overlap between Gaussians that are offset by some amount accounts for the error term. The smallest possible offset is equal to the period $\pi$ of the $\cos(2q)$ potential, and so the error estimate in our expression for $\int_0^{2\pi} dq \, \overline{\chi_k(q)} \chi_k(q)$ follows from taking $d = \pi$ in Eq. (151). Finally, for later use we remind the reader that for our model the parameter $\lambda$ is proportional to $\xi/\ell$, where the correlation length $\xi$ was defined in Eq. (136).

### E.3 Calculating the Berry phase $\gamma_k$

We now calculate the Berry phase $\gamma_k$ in Eq. (51) using properties of the eigenstates of our effective zero mode Hamiltonian $H_{\text{eff}}$. To simplify the notation we denote $|\psi_k(B, 0)\rangle$ by $|\psi_k\rangle$ in what follows.

To start, we note that $\langle \psi_k | \tilde{p} | \psi_k \rangle = \langle \psi_k | p | \psi_k \rangle + \delta$, and so we focus on evaluating $\langle \psi_k | p | \psi_k \rangle$. For this matrix element we have

$$\begin{aligned}
\langle \psi_k | p | \psi_k \rangle &= -i \int_0^{2\pi} dq \, \overline{\psi_k(q)} \frac{d}{dq} \psi_k(q) \\
&= -\delta \int_0^{2\pi} dq \, |\chi_k(q)|^2 - i \int_0^{2\pi} dq \, \overline{\chi_k(q)} \frac{d}{dq} \chi_k(q) \,, 
\end{aligned} \tag{152}$$

where we remind the reader that $\psi_k(q) = e^{-i\delta q} \chi_k(q)$. We then use the approximate form (149) of the wave functions $\chi_k(q)$ at large $\ell$ (more precisely, at large $\lambda$) to find that

$$\int_0^{2\pi} dq \, |\chi_k(q)|^2 = 1 + O(e^{-\frac{\pi^2}{8}\frac{\ell}{\xi}}) \tag{153}$$

and

$$-i \int_0^{2\pi} dq \, \overline{\chi_k(q)} \frac{d}{dq} \chi_k(q) = O(e^{-\frac{\pi^2}{8}\frac{\ell}{\xi}}) \,, \tag{154}$$

where the error terms on these estimates come from the calculation of the overlap of two shifted Gaussians (recall Eq. (151)). Therefore our final result is that

$$\langle \psi_k | p | \psi_k \rangle = -\delta + O(e^{-\frac{\pi^2}{8}\frac{\ell}{\xi}}) \,, \tag{155}$$

---

[7]In our problem the original wave functions are defined for $q \in [0, 2\pi)$. This explains our extra factor of $\sqrt{2}$ as compared with Ref. [86], where the wave functions were normalized for integration over one period of the periodic potential (which is $\pi$ in our case).

and so we find that the Berry phase $\gamma_k$ is given (within our variational approximation) by

$$\gamma_k = -k\pi + O(e^{-\frac{\pi^2}{8}\frac{\ell}{\xi}}) \, . \tag{156}$$

The most interesting aspect of our result for $\gamma_k$ is that, for $\ell \gg \xi$, the Berry phase is equal to the *topological* value

$$\gamma_{k,\text{top}} = -k\pi \, , \tag{157}$$

up to corrections that are exponentially small in the length $\ell$ (which is also the separation between the fractional quasiparticles at the ends of the FM region). These exponentially small corrections to topological Berry phases are always expected in finite size systems, but they are very rarely calculated explicitly. Our ability to capture these corrections here is a significant demonstration of the power of our variational method.

## F  Generalization to $m \neq 1$

In this appendix we briefly explain the generalization of our results to the fractional case of $m > 1$ (i.e., a domain wall configuration at the boundary of a fractional quantum spin Hall system). Recall from Appendix C that in the $m = 1$ case we were able to precisely construct bosonized fermion operators $R(s)$ and $L(s)$ that obey the correct anticommutation relations of fermion field operators. In contrast to that result, in the $m > 1$ case we are not aware of a precise construction of bosonized fermion operators $R(s)$ and $L(s)$ that exactly obey the correct anticommutation relations. One possible guess in this case is to define $R(s)$ and $L(s)$ via

$$R(s) \;\; = \;\; \frac{e^{i\delta\frac{\pi x}{\ell}}}{\sqrt{2\ell}} : e^{-im\phi_\uparrow(s)} : \, , \tag{158a}$$

$$L(s) \;\; = \;\; \frac{e^{i\delta\frac{\pi x}{\ell}}}{\sqrt{2\ell}} : e^{im\phi_\downarrow(s)} : \, . \tag{158b}$$

With these definitions one still finds that $\{R(s), R(y)\} = 0$ and $\{L(s), L(y)\} = 0$. However, the other anticommutators no longer exactly match the expected answer for fermionic operators. For example, in the limit of $\epsilon \to 0$, $\{R(s), R^\dagger(y)\} \neq \delta_{2\ell}(x-y)$ but is instead equal to some more complicated distribution.[8] Heuristically, the deviation between $\{R(s), R^\dagger(y)\}$ and $\delta_{2\ell}(x-y)$ is due to a short length scale of the strongly interacting system above which interacting fermion systems develops topological order, which we can identify as the ultraviolet cutoff $\epsilon$ in the mode expansion (71) of the boson fields.

Because of this issue, in this appendix only we adopt a less precise (but commonly used) definition of the bosonized fermion operators. Specifically, we define $R(s)$ and $L(s)$ via

$$R(s) \;\; \sim \;\; \frac{1}{\sqrt{2a}} e^{-im\phi_\uparrow(s)} \, , \tag{159a}$$

$$L(s) \;\; \sim \;\; \frac{1}{\sqrt{2a}} e^{im\phi_\downarrow(s)} \, , \tag{159b}$$

where we have not used any normal-ordering prescription, and where we used the ultraviolet cutoff $a$ (instead of the infrared cutoff $\ell$) to obtain the correct dimensions. Loosely speaking, using this definition we have $\{R(s), R^\dagger(0)\} = 0$ if $x \neq 0$, and $\{R(s), R^\dagger(0)\} \sim 1/a \to \infty$, similar to a $\delta$-function.

---

[8]This fact about the bosonized fermion operators in the fractional case does not seem to be widely known. At least, we are not aware of any discussion of it in the literature.

We again carry out a variational calculation using a trial state $|\Psi\rangle = |\psi\rangle \otimes |0, \zeta\rangle$, where $\zeta = (\zeta_1, \zeta_2, \dots)$ is again chosen so that the expectation value of the magnetic field term in the state $|\Psi\rangle$ is extensive. In particular, in this case we find that

$$\langle \Psi | R^\dagger(s) L(s) | \Psi \rangle \sim \frac{1}{2} f_m(x; a; \zeta) \langle \psi | e^{i2mq} | \psi \rangle \,, \tag{160}$$

where the function $f_m(x; a; \zeta)$ is given by

$$f_m(x; a; \zeta) = \frac{1}{a} e^{i \frac{\pi m x}{\ell}} e^{\frac{i}{m} \sum_{n=1}^\infty \frac{e^{-\epsilon n}}{n} \sin(2\kappa_n x)}$$
$$\times e^{-\sum_{n=1}^\infty \frac{e^{-\epsilon n}}{n} e^{-2(\eta' + \zeta_n)}(1 + \cos(2\kappa_n x))} \,, \tag{161}$$

where now

$$e^{-2\eta'} = K' \equiv mK \,. \tag{162}$$

Since the first line in Eq. (161) is equal to $1/a$ times a pure phase factor (i.e., a complex number of unit modulus), we find that by performing the summation in the exponent of the last factor of Eq. (161) with the same variational scheme in Appendix D,

$$f_m(x; a; \zeta) = \Delta_Z^{K'} a^{K'-1} f_{m,0}(x; a; \zeta), \tag{163}$$

where $f_{m,0}(x; a; \zeta)$ is an order one quantity. This result is very similar to the $m = 1$ case from Appendix D, with the important difference that $K$ is replaced by $K' = mK$.

In this way we find that $|\psi\rangle$ should again be chosen to be the ground state of an effective zero mode Hamiltonian, and in this case this zero mode Hamiltonian takes the form

$$H_{\text{eff}} = \alpha \tilde{p}^2 - \beta \cos(2mq) \,. \tag{164}$$

Following the self-consistency relation in Appendix D, we have

$$\alpha = \frac{\nu K' \pi}{2\ell}, \tag{165a}$$

$$\beta \sim B^{\frac{2}{2-K'}} a^{\frac{2K'-2}{2-K'}} \ell. \tag{165b}$$

We see that we again have $\alpha \propto \frac{1}{\ell}$ and $\beta \propto \ell$, and so we again have

$$\lambda^2 = \frac{\beta}{\alpha} \propto \ell^2 \,. \tag{166}$$

The main difference between the analysis in this case and the analysis in the $m = 1$ case is that $H_{\text{eff}}$ (and the full domain wall Hamiltonian $H(a)$) have a $\mathbb{Z}_{2m}$ symmetry instead of a $\mathbb{Z}_2$ symmetry. This symmetry is generated by the operator $e^{i\frac{\pi p}{m}}$, and it can again be related to the conservation of the parity of the spin $S$ in the FM region. Indeed, in this case we have

$$S = \frac{1}{2\pi} \int_0^\ell ds \left[ \partial_s \phi_\uparrow(s) - \partial_s \phi_\downarrow(s) \right] = \frac{\tilde{p}}{m} \,, \tag{167}$$

and the Hamiltonian commutes with $e^{i\pi S} = e^{i\frac{\pi \tilde{p}}{m}}$. The Hilbert space of the model breaks up into sectors labeled by the different eigenvalues of the $\mathbb{Z}_{2m}$ symmetry operator, and for our convenience we choose to label the different sectors by their eigenvalue of $e^{i\pi S} = e^{i\frac{\pi \tilde{p}}{m}}$, which involves the shifted momentum operator $\tilde{p}$.

Consider the sector of the Hilbert space characterized by $e^{i\frac{\pi \tilde{p}}{m}} = e^{i\frac{\pi k}{m}}$, where $k$ takes on one of the $2m$ values in the set $\{-m + \delta, \dots, -1 + \delta, \delta, \dots, m-1+\delta\}$. Our variational approximation for the ground state of $H(a)$ in this sector is the trial state $|\Psi_k\rangle = |\psi_k\rangle \otimes |0, \zeta\rangle$, where

$|\psi_k\rangle$ should be chosen to be the ground state of $H_{\text{eff}}$ in the sector with $e^{i\frac{\pi\tilde{p}}{m}} = e^{i\frac{\pi k}{m}}$. By again exploiting the connection to the Mathieu's equation,[9] we find that
$\langle q|\psi_k\rangle = \psi_k(q) = e^{-i\delta q}\chi_k(q)$, where now the function $\chi_k(q)$ should be chosen to be the eigenfunction in the lowest band of the operator $-\frac{d^2}{dq^2} - \lambda^2 \cos(2mq)$ that also satisfies the periodicity condition

$$\chi_k(q + \tfrac{\pi}{m}) = e^{ik\frac{\pi}{m}}\chi_k(q) . \tag{168}$$

All of our previous results can now be carried over to this case. The only difference is that there are now small changes in the asymptotic formula for the width $|W_0(\lambda)|$ of the lowest band of the Mathieu operator, and the approximate form of the eigenfunction $\chi_k(q)$ in the tight-binding regime of large $\lambda$. These quantities are now given by

$$|W_0(\lambda)| = m^2 \frac{2^{\frac{19}{4}}}{\pi^{\frac{1}{2}}} \left(\frac{\lambda}{m}\right)^{\frac{3}{2}} e^{-\frac{\lambda}{m}\sqrt{8}} \left[1 + O(\lambda^{-\frac{1}{2}})\right] , \tag{169}$$

and

$$\chi_k(q) = \frac{1}{\sqrt{2m}} \sum_{n\in\mathbb{Z}} e^{ikn\frac{\pi}{m}} \chi_0(q - n\tfrac{\pi}{m}) , \tag{170}$$

where now

$$\chi_0(q) = \left(\frac{m\lambda\sqrt{2}}{\pi}\right)^{\frac{1}{4}} e^{-\frac{m\lambda}{\sqrt{2}}q^2} . \tag{171}$$

Note that the new factors of $m$ in $|W_0(\lambda)|$ can be understood from the expression for $|W_0(\lambda)|$ at $m = 1$ by making the change of variables $q' = mq$ in the Mathieu's equation with $m \neq 1$. Also, the factor of $1/\sqrt{2m}$ in the expression for $\chi_k(q)$ is again present to ensure approximate normalization when integrated over the interval $[0, 2\pi)$, which is $2m$ times larger than the period $\pi/m$ of $\cos(2mq)$.

Using these new formulas we again predict (in the tight-binding regime) an exponentially small splitting between the ground state energies $E_k$ of $H(a)$ in sectors with different values of $k$,

$$|E_{k_1} - E_{k_2}| \lesssim \text{constant} \times e^{-\frac{\ell}{\xi_m}} , \tag{172}$$

where the new correlation length $\xi_m$ is of the same order as the correlation length in the $m = 1$ case. Finally, we find that the Berry phase $\gamma_k$ associated with the full $2\pi$ rotation of the in-plane magnetic field is given approximately by

$$\gamma_k = -k\frac{\pi}{m} + O(e^{-\frac{\ell}{\xi_m}\frac{\pi^2}{8}}) . \tag{173}$$

The main difference compared to the integer case is the presence of the factor of $1/m$, indicating a fractional value for the Berry phase. We again find exponential suppression of the corrections to this topological value. One important point for this Berry phase calculation is that we now choose the phase of the state $|\psi_k(B, \tau)\rangle$ according to the formula

$$|\psi_k(B, \tau)\rangle = e^{i\frac{\tau k}{m}} e^{-i\frac{\tau\tilde{p}}{m}} |\psi_k(B, 0)\rangle , \tag{174}$$

and this choice will ensure that the states are single-valued along the path that we take through the parameter space. In particular, with this choice we will again have $|\psi_k(B, \tau + \pi)\rangle = |\psi_k(B, \tau)\rangle$.

---

[9]Actually, the standard form of Mathieu's equation has the potential $\cos(2q)$, which has a period of $\pi$. In our case we instead have $\cos(2mq)$, with a period of $\pi/m$, but it is a simple matter to take this rescaling of the period into account in our analysis.

# G   Exact two-fold degeneracy of the domain wall model at $\delta = 1/2$

Our main interest in this paper is domain wall configurations in which the central FM region is surrounded by two SC regions with *opposite* signs of the superconducting mass $\Delta(s)$. In this case, the central FM region is described by our domain wall model with the parameter value $\delta = \frac{1}{2}$. In this appendix we show that in this situation the domain wall Hamiltonian $H(a)$ has an *exact* two-fold degeneracy of all of its eigenstates (and this holds for any value of the integer $m$). We explain this symmetry structure in the particular case that the in-plane magnetic field **B** points along the positive $x$-axis, as the Hamiltonian with a rotated **B** is unitarily equivalent to this case (and so the structure of the energy spectrum will be the same).

We start by noting that, since $H(a)$ commutes with $e^{i\frac{\pi p}{m}}$, it also commutes with the $\mathbb{Z}_2$ symmetry operator $\Gamma_1 = e^{i\pi p} = (e^{i\frac{\pi p}{m}})^m$, which satisfies $\Gamma_1^2 = 1$ since $p$ has integer eigenvalues. For $m = 1$ this is the fermion parity symmetry. Next, we identify a second operator $\Gamma_2$ that (i) commutes with $H(a)$, (ii) squares to the identity, $\Gamma_2^2 = 1$, and (iii) *anticommutes* with $\Gamma_1$, $\{\Gamma_1, \Gamma_2\} = 0$. The existence of two operators $\Gamma_1$ and $\Gamma_2$ with these properties implies the two-fold degeneracy of all eigenstates of $H(a)$. Indeed, if $|\Psi\rangle$ is an eigenstate of $H(a)$ with $\Gamma_1|\Psi\rangle = |\Psi\rangle$, then these properties imply that $|\Psi'\rangle = \Gamma_2|\Psi\rangle$ is an eigenstate of $H(a)$ with the same energy as $|\Psi\rangle$, but with $\Gamma_1|\Psi'\rangle = -|\Psi'\rangle$.

We define the operator $\Gamma_2$ by its action on the operators $q, p, b_n$, and $b_n^\dagger$ that appear in the mode expansions of the bosonic fields $\varphi(s)$ and $\vartheta(s)$ in our model. As we mentioned above, we also choose $\Gamma_2$ to be anti-unitary. We define $\Gamma_2$ in such a way that it squares to the identity operator,

$$\Gamma_2^2 = 1 \, , \tag{175}$$

and we define its actions on $q, p, b_n$, and $b_n^\dagger$ as:

$$\Gamma_2 q \Gamma_2 = q \, , \tag{176a}$$

$$\Gamma_2 p \Gamma_2 = -p - 1 \, , \tag{176b}$$

$$\Gamma_2 b_n \Gamma_2 = b_n \ \forall \ n \, , \tag{176c}$$

$$\Gamma_2 b_n^\dagger \Gamma_2 = b_n^\dagger \ \forall \ n \, . \tag{176d}$$

Therefore, $\Gamma_2$ only acts nontrivially on $p$. However, it can also act on other expressions by complex conjugation since it is anti-unitary. We also note that $\Gamma_2 \tilde{p} \Gamma_2 = -\tilde{p}$, where $\tilde{p} = p + \frac{1}{2}$ at $\delta = \frac{1}{2}$. With these definitions one can easily see that $\Gamma_2 \varphi(s) \Gamma_2 = \varphi(s)$ and $\Gamma_2 \vartheta(s) \Gamma_2 = -\vartheta(s)$. From Eq. (59) of the main text, this $\Gamma_2$ operator is precisely the antiunitary time-reversal symmetry $\tilde{\mathcal{T}}$.

These relations in turn imply that $\Gamma_2 \phi_\uparrow(s) \Gamma_2 = \phi_\downarrow(s)$ and $\Gamma_2 \phi_\downarrow(s) \Gamma_2 = \phi_\uparrow(s)$. Finally, these relations imply that the bosonized fermion operators $R(s)$ and $L(s)$ satisfy

$$\Gamma_2 R(s) \Gamma_2 = L(s) \, , \tag{177a}$$

$$\Gamma_2 L(s) \Gamma_2 = R(s) \, , \tag{177b}$$

and so we find that $\Gamma_2$ does indeed commute with the domain wall Hamiltonian $H(a)$ at $\delta = \frac{1}{2}$.

Finally, we investigate the interplay between $\Gamma_2$ and $\Gamma_1$. We have

$$\Gamma_2 \Gamma_1 \Gamma_2 = \Gamma_2 e^{i\pi p} \Gamma_2 \, , \tag{178}$$

$$= e^{-i\pi(-p-1)} \, ,$$

$$= -e^{i\pi p} \, ,$$

$$= -\Gamma_1 \, , \tag{179}$$

and so $\Gamma_2$ anticommutes with $\Gamma_1$. This completes our demonstration of the three properties of $\Gamma_2$ that we stated above. As we mentioned above, this then implies an exact two-fold degeneracy of all of the eigenstates of the domain wall Hamiltonian $H(a)$.

# H  Ground state degeneracy from the perspective of the 't Hooft anomaly

In this Appendix we show that the ground state degeneracy due to the Majorana pair in each corner can be viewed as a consequence of a mixed 't Hooft anomaly between the generalized time-reversal symmetry $U_{s,\pi}\mathcal{T}$ and fermion parity symmetry $U_{s,2\pi}$.

We begin with the partition function of the corner region in terms of the boson fields $\vartheta$ and $\varphi$ (we set $v = 1$ and keep a generic $m$), given by $Z = \int \mathcal{D}\varphi \mathcal{D}\vartheta e^{iS}$, where

$$S[\vartheta, \varphi] = \frac{1}{2\pi} \int dt \int_0^\ell ds \left[ 2\partial_x(\vartheta + \alpha)\partial_t\varphi - K'(\partial_x\vartheta)^2 - \frac{1}{K'}(\partial_x\varphi)^2 + 2\pi b \cos(2\varphi) \right], \quad (180)$$

subject to the spatial boundary condition

$$\vartheta(\ell) - \vartheta(0) = \left( p + \frac{1}{2} \right) \frac{\pi}{m}, \quad p \in \mathbb{Z}. \quad (181)$$

The parameter $\alpha$ is rather unusual and absent from most literature on bosonization, which we will explain and determine shortly. Recall that the first term arises from the insertion of complete sets of conjugate coherent states $|\varphi\rangle$ and $|\pi\rangle \equiv |\partial_x(\vartheta + \alpha)\rangle$, which gives the matrix element

$$\prod_x \langle \varphi(x, t + dt)|\pi(x, t)\rangle\langle \pi(x, t)|\varphi(x, t)\rangle = \exp\left[ dt \int \frac{i\partial_x(\vartheta(s) + \alpha(s))\partial_t\varphi(s)}{\pi} ds \right]. \quad (182)$$

Indeed, it is straightforward to verify that $[\varphi(s), \pi(s')] = i\pi\delta(s - s')$.

In the above we have used

$$\prod_x \langle \varphi(x, t)|\pi(x, t)\rangle = \exp\left[ \int \frac{i\partial_x(\vartheta(s) + \alpha(s))\varphi(s)}{\pi} ds \right]. \quad (183)$$

and from this the parameter $\alpha(s)$ can be determined by noticing the Hilbert space constraint of the compactification $\varphi(s) \sim \varphi(s) + 2\pi m$. This requires that

$$\int_0^\ell \frac{dx}{\pi} \partial_x(\theta + \alpha) \in \frac{\mathbb{Z}}{m}. \quad (184)$$

Given the spatial boundary condition Eq. (181), we can choose

$$\alpha = \frac{\Theta x}{2m}, \quad \Theta = \pi. \quad (185)$$

Note that this procedure is essentially the same as the one adopted in Eq. (48).

After integrating the $\vartheta$ field, this leads to the action

$$S[\varphi] = \int \frac{d^2x}{2\pi} \left[ \frac{\Theta\partial_t\varphi}{m} - \frac{(\partial_\mu\varphi)^2}{K'} + 2\pi b \cos(2\varphi) \right]. \quad (186)$$

Notice that compared to the usual sine-Gordon model, we have an additional term with $\Theta = \pi$. After integrating over $x \in [0, \ell]$, this is precisely a $\Theta$-term in a 1d quantum field theory.

The partition function $Z = \int e^{iS}$ has two symmetries, a generalized time-reversal $\tilde{\mathcal{T}}$ under which $\varphi \to \varphi$, $t \to -t$, and a translation $\varphi \to \varphi + \pi$ (for $m = 1$ this is fermion parity $U_{s,2\pi}$). In particular, the former symmetry is only realized at $\Theta = 0, \pi$ in the presence of periodic temporal boundary conditions. As pointed out in Ref. [90], such the theory $\Theta = \pi$ admits a

't Hooft anomaly between the two symmetries. To this end, we couple the spin up and down fermions with a gauge field $\pm A^s$, via $\partial \varphi \to \partial \varphi - A^s$, and we show that gauge invariance and time-reversal symmetry are incompatible.

After integrating out spatially oscillatory modes, we have

$$S[q,A^s] = \int \frac{dt}{2\pi}\Big[\Theta(\dot{q} - A^s/m) - \frac{(m\dot{q} - A^s)^2}{K'} + 2\pi\beta\cos(2mq)\Big], \quad \Theta = \pi. \tag{187}$$

With the cosine term, the gauge group is lowered from U(1) to $\mathbb{Z}_{2m}$. Indeed this partition function

$$Z[A^s] = \int dq\, e^{S[q,A^s]} \tag{188}$$

is gauge invariant, including the large gauge transformation

$$\int dt\dot{q} \to \int dt\dot{q} + 2\pi, \quad \int dtA^s \to \int dtA^s + 2\pi m. \tag{189}$$

However, in doing so, we have introduced a 1d Chern-Simons counter-term $\sim \int dtA^s$, which necessarily breaks time-reversal symmetry, since $A^s$ is odd under time reversal.

We can alternatively keep time-reversal symmetry, by taking a different way of coupling to the gauge field

$$S[q,A^s] = \int \frac{dt}{2\pi}\Big[\Theta\dot{q} - \frac{(m\dot{q} - A^s)^2}{K'} + 2\pi\beta\cos(2mq)\Big], \quad \Theta = \pi. \tag{190}$$

However, this theory is not gauge invariant under the large gauge transformation above. A simple analysis shows that the partition function

$$Z[A^s] \to -Z[A^s] \tag{191}$$

under such a transformation. Here the incompatibility of $\mathbb{Z}_{2m}$ and time-reversal of the quantum theory is characteristic of a 't Hooft anomaly.

In general, the ground state degeneracy due to the 't Hooft anomaly can be proven by contradiction. Suppose there is a unique ground state, and then due to time-reversal symmetry of the partition function $Z[A_s]$, the ground state must carry zero charge under the gauge field, since the (temporal) gauge field $A^s$ is odd under time-reversal. However, if so, the ground state path integral could not admit a gauge anomaly, since being charge neutral it would not respond to any gauge transformation. Therefore, the ground state must be degenerate.

In this special case of $m = 1$, the symmetry properties of $Z[A^s]$ from Eq. (191) can be captured by [90]

$$Z[A^s] \sim \exp\left(i\int dtA^s/2\right) + \exp\left(-i\int dtA^s/2\right), \tag{192}$$

which indicates that the ground state is two-fold degenerate in the absence of the background gauge field. Each ground state carries a fractional charge $\pm\frac{1}{2}$, and therefore, the gauge group is represented projectively, or equivalently as a double cover. While classically the $\mathbb{Z}_2$ time-reversal symmetry and the $\mathbb{Z}_2$ gauge symmetry combines to $\mathbb{D}_4$, at a quantum level the symmetry group is $\mathbb{D}_8$.

Recalling that spin-1/2 fermions are charged $\pm 1$ objects under $A_s$, we conclude that the two ground states have spin $S = \pm\frac{1}{4}$. This indeed agrees with the results from the main text using (41)

$$S = \frac{1}{2}\int_0^\ell ds\frac{\partial_x\vartheta}{\pi} = \frac{1}{4} \mod \frac{1}{2}. \tag{193}$$

# Acknowledgements

We thank J.-H. Chu, P. Hirschfeld, A. Jahin, A. Tiwari, F. Zhang and X.-X. Zhang for useful discussions.

**Funding information** M.F.L. acknowledges the support of the Kadanoff Center for Theoretical Physics at the University of Chicago. M.F.L. is also supported by the Simons Collaboration on Ultra-Quantum Matter, which is a grant from the Simons Foundation (651440). M.C. acknowledges support from NSF under award number DMR-1846109 and the Alfred P. Sloan foundation. Y.W. is supported by startup funds at the University of Florida and by NSF under award number DMR-2045871.

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
