# Peer review of "Symmetry-protected gates of Majorana qubits in a high-Tc superconductor platform"

_SciPost Physics, doi:SciPost Phys. 11, 086 (2021)_

## Round 1 · Referee Report · Anonymous (Referee 1) · 2021-6-22

Report

The authors study an effective edge theory of a heterostructure consisting of a QSH insulator and a d-wave superconductor. The analyze the effective symmetries of the theory and the existence of corner Majorana zero modes in the presence of domain walls caused by local Zeeman fields. Then, they propose a way to implement symmetry-protected quantum gates via the rotations of the Zeeman field and its effect on the position of the MZM. Interestingly they point out that non-Clifford rotations can be implemented via a "slice of pie" contour in the (Bx, By) plane. The formalism used allow generalization to the fractional case, and several lengthy appendices quench the thirst for technical details.

This work is interesting and timely given recent interest on heterostructures of the type studied here. Indeed, the authors point out a few explicit material examples (WTe and BSCCO), although their theoretical analysis is quite far from realistic samples. On the technical side, the zero-mode analysis is carried out with lucidity and using known techniques, which leaves little doubt as to the correctness of their results. In fact, the effective edge theory is that of a QSH edge subject to local Zeeman and s-wave pairing (with possible sign changes in Delta inherited by the parent d-wave pairing), which is familiar territory.

I find that this work meets one of the expectations for acceptance in SciPost (namely, "Open a new pathway in an existing or a new research direction, with clear potential for multipronged follow-up work") and has the clear potential to meet all of the required general acceptance criteria. Thus, I believe that this work should be eventually be published in SciPost Physics. I have some remarks on the clarity of the manuscript which I think deserve further edits and review before publication.

First, I find that the setup and coordinate system should be better described.

• How exactly does the setup in Fig. 1 avoid the problem of single-particle tunneling from nodal gapless states in the parent superconductor?
• What is the coordinate system used? Is x a coordinate that parametrizes the position along the edge in Fig.1 (including the corner), or is the (x,y) coordinate system fixed with respect to the sample? It is not always clear to follow the conventions adopted throughout the text, especially when it comes to the field labeling.

Second, the simplicity and robustness of the proposal are somewhat oversold and should be discussed more critically.

• Most importantly, I do not understand the robustness of the "pie slice" protocol in Fig. 4. Even if all required symmetry conditions are satisfied, it seems to me that a small error in implementing the angle tau0 will result in a linear error in the final phase gate of Eq. 60. So how can the Berry phase be robust to small deformations of the arc? Practically speaking, what is the advantage with respect to a poor-man's rotation implemented via the timing of an interaction between MZMs?
• The effective time-reversal symmetry T_tilde seems to be crucial for protected operations. What are examples of perturbations that break this symmetry? Can they be expected in "real" devices?
• The entire scheme seems rely on the absence of disorder and on the presence of several crystalline symmetry. Can it survive against, say, irregularities or non-uniformity in the sample shape? For instance, it seems to me that a certain relative orientation between the QSH insulator and the d-wave superconductor must be required to avoid the interaction between the nodal gapless excitations, but this aspect, while mentioned, is not clearly illustrated and looks worryingly fragile.

Finally, the clarity of the presentation would benefit from one thorough proof-reading. Overall, the style and length of the work are appropriate and pleasant, but the notation is inconsistent or hard to trace at times, an

---

## Round 1 · Referee Report · Anonymous (Referee 2) · 2021-6-25

Report

In their work, the authors propose a way to perform arbitrary quantum gates by manipulating Majorana corner states in a second-order topological superconductor. The starting point is a time-reversal invariant second-order topological superconductor as proposed in Refs. [61,62] of the manuscript. In such a system, each corner of a rectangular sample hosts a Kramers pair of Majorana corner states. When an additional in-plane field is applied, time-reversal symmetry is broken and the Kramers partners can split in space. Importantly, however, the presence of an additional emergent time-reversal symmetry still prevents the two Majoranas at a given corner from hybridizing. The authors show that changing the magnetic field along certain closed paths in parameter space gives rise to non-Abelian Berry phases that realize non-trivial rotations in the degenerate ground state manifold accompanying the Majorana corner states. It is proposed to use these rotations to perform quantum gates on qubits encoded within the degenerate ground state manifold.

The manuscript addresses a timely topic, is well-written, and appears to be technically sound. There are, however, several points that I hope the authors can comment on further. In particular, I have some concerns regarding the experimental feasibility and stability of the proposed manipulations. Furthermore, there are also some technical points that would benefit from an extended discussion.

Explicitly, the points I would like to raise are the following:

1. While the authors advertise their proposal as experimentally feasible and advantageous compared to several other computational schemes, there are a number of issues that might arise in an experimental realization of the proposed setup:

1.1. The authors give a detailed discussion of the symmetries required to protect the Berry phases under consideration. However, these symmetry constraints appear to be very stringent (see, e.g., the conditions (1)-(3) listed on pages 18 and 19 of the manuscript) and I am wondering how well they will be fulfilled in any experimental implementation of the proposed setup. In particular, it seems that an out-of-plane component of the Zeeman field would immediately break the emergent time-reversal symmetry required to protect the proposed phase gates. Given the extensive (and experimentally non-trivial) manipulations of the in-plane Zeeman fields required to achieve the proposed quantum gates, it seems extremely challenging to completely avoid the accidental presence of a small out-of-plane component of the Zeeman field. This is even more so, since the Zeeman fields have to be local and are assumed to be non-zero only at a given corner where the manipulation is supposed to take place. But how is this possible with only in-plane B-fields which also need to satisfy div B=0?

1.2. The authors assume perfect experimental control over the applied Zeeman fields. In particular, it is assumed that the path along which the Zeeman field evolves in the (B_x,B_y) plane is always a closed one. However, what if---by accident---one does not end up at the same point in parameter space from which one has started, i.e., the path in parameter space is not closed? If the experimental control of the Zeeman field is not perfect, such errors could easily go unnoticed. There could also, for example, be a danger of ‘overshooting’ the magnetic field and performing a 2\pi\pm\epsilon rotation instead of an exact 2\pi rotation (in case of a full braid). How would this affect the results presented in the manuscript, and are there ways to detect, remedy, or avoid such errors?

1.3. One should mention that a d-wave superconductor by itself already presents a highly non-trivial state of matter. It would be interesting to see if similar results could also be obtained from a time-reversal invariant topological insulator engineered from conventional ingredients only (see, e.g., Phys. Rev. B 102 (19), 195401).

2. When the model is studied in bosonic language, the considered situation is slightly different from the initial situation studied in the fermionic case, see Fig. 2. In the latter case, the length l depends on the strength of the Zeeman field and goes to zero as the Zeeman field goes to zero. In the model that is studied in the bosonic language, however, the length l is fixed and stays finite at all times. This is crucial for the expression for the level spacing given in Eq. (46) and as such also for stability of the proposed ‘half-moon’ and ‘slice-of-pie’ contours where the Zeeman field goes through zero at some point. The authors should comment on the differences between the two models studied in the fermionic and the bosonic language. Do these differences affect the results that are obtained and if so, in what way? Which situation is closer to the one actually encountered in an experimental realization of the model?

3. In the case of the ‘half-moon’ and ‘slice-of-pie’ contours, the Zeeman field goes to zero at some point of the process. In this case, the gap separating the ground state manifold from the excited states is given by the finite-size level spacing, see Eq. (46). I would expect that this gap can become quite small, which would then again put strong constraints on the adiabaticity of the process. Can the authors give an estimate on the energy scales involved in a realistic implementation of their scheme, and specify what this means for the adiabaticity condition of the process?

4. The intuition behind the ‘braiding’ process via a 2pi rotation of the in-plane Zeeman field could be made clearer. As far as I understand, there is no actual spatial exchange of the two Majoranas. Rather, the process seems to be closely connected to the one discussed in arXiv:1803.02173 (Ref. [58] in the manuscript). What is the intuition behind calling such a process an actual ‘braiding’ process? It would greatly benefit the accessibility of the manuscript if the authors could add some additional explanations here.

5. It should be emphasized more clearly that some of the proposed manipulations (in particular, the arbitrary phase gates via ‘slice-of-pie’ contours) do not correspond to quasiparticle braiding processes. While the introduction correctly states that it is fundamentally impossible to realize non-Clifford gates by braiding Majoranas, this distinction is somewhat blurred in the main part of the manuscript. This is also due to the fact that even the single exchange (‘half-moon’ contour) does not correspond to a standard exchange of Majorana quasiparticles as one would typically have in mind when talking about topological quantum computation by non-abelian anyons. As the magnetic field goes to zero during the exchange process, the two Majoranas strongly overlap and the inherent topological protection accompanying a set of well-separated Majoranas is lost. What one is left with is merely a symmetry-protected non-Abelian Berry phase that could emerge also in any other two-level system (not based on anyonic quasiparticles) in the presence of suitable symmetries. Overall, the processes discussed in the manuscript at hand seem to be closely related to holonomic quantum computation. The authors should comment on these connections more transparently.

6. The results on the parafermionic case come with some unresolved problems such as unwanted dynamical phases. While this is correctly pointed out in the main text, I recommend that the corresponding part of the abstract should be phrased more carefully. In particular, the sentence “Our analysis naturally includes interaction effects and can be generalized to cases with fractional bulk excitations” tends to oversell the results on the parafermionic case.

7. Finally, I have a few more detailed remarks regarding specific parts of the manuscript:

7.1. Fig. 1: I do not understand the meaning of the red oval in the drawing. Does it represent a path in parameter space (=direction of magnetic field)?
7.2. Page 6, in between Eqs. (2) and (3): I am not sure I understand what is meant by the remark ‘We choose x to be along the diagonal direction.’ From the equations, I would assume x to be an edge coordinate running along the edge of a finite sample. It would be helpful to indicate the coordinate axes in Fig. 1, for example.
7.3. In Eq. (9): x_0 is not defined. Should it be x_-?
7.4. On page 7, after Eq. (9), it reads: “[…] there exist a pair of MZMs separated by a length l, the length of the region where |\Delta|=B.” Should it be |\Delta|<B?
7.5. Eq. (15): Is the complex conjugation on the RHS correct? It is already included in the definition of P.
7.6. Page 8, after Eq. (16): I believe that \bar{C} is not defined.
7.7. Eq. (18b): It should be x’ instead of x in the second argument on the LHS.
7.8. Eqs. (55), (56), and (57): I do not understand the difference between the \psi in the bra and \Psi in the ket. Is it a different state?
7.9. Page 18, condition (2): I believe there is a typo – the same condition Ba>>(a/l)^{2-K} is given twice. The same question also applies to page 19, again condition (2).
7.10. Eq. (68): The use of \tau_0 is somewhat unfortunate here, since the same symbol is also used for the zeroth Pauli matrix in particle-hole space [see, e.g., Eq. (10)].
7.11. Eq. (70): I do not understand the meaning of (and the difference between) k_a, k_b, k_x, and k_y.
7.12. Page 23, paragraph after Eq. (71): H_B is not defined. Should it be H_Z?

---

## Round 2 · Referee Report · Anonymous (Referee 1) · 2021-8-25

Report

I would like to thank the Authors for their reply and the edits to the manuscript. My questions and requests for clarifications have been answered satisfactorily, and are adequately reflected in the minimal changes made to the manuscript.

Given that the quality of research and validity of the manuscript were already judged to be of high level by both Referees in the first round of review, in my opinion this work is ready for publication in SciPost Physics. It meets the acceptance criteria as detailed in my first report.

---

## Round 2 · Author Response

Below we address the comments from both referees.

Referee 1

The authors study an effective edge theory of a heterostructure consisting of a QSH insulator and a d-wave superconductor. The analyze the effective symmetries of the theory and the existence of corner Majorana zero modes in the presence of domain walls caused by local Zeeman fields. Then, they propose a way to implement symmetry-protected quantum gates via the rotations of the Zeeman field and its effect on the position of the MZM. Interestingly they point out that non-Clifford rotations can be implemented via a "slice of pie" contour in the (Bx, By) plane. The formalism used allow generalization to the fractional case, and several lengthy appendices quench the thirst for technical details. This work is interesting and timely given recent interest on heterostructures of the type studied here. Indeed, the authors point out a few explicit material examples (WTe and BSCCO), although their theoretical analysis is quite far from realistic samples. On the technical side, the zero-mode analysis is carried out with lucidity and using known techniques, which leaves little doubt as to the correctness of their results. In fact, the effective edge theory is that of a QSH edge subject to local Zeeman and s-wave pairing (with possible sign changes in Delta inherited by the parent d-wave pairing), which is familiar territory. I find that this work meets one of the expectations for acceptance in SciPost (namely, "Open a new pathway in an existing or a new research direction, with clear potential for multipronged follow-up work") and has the clear potential to meet all of the required general acceptance criteria. Thus, I believe that this work should be eventually be published in SciPost Physics. I have some remarks on the clarity of the manuscript which I think deserve further edits and review before publication.

—We thank the referee for their positive evaluation of our work. In particular for commending our work for potentially “opening a new pathway… with clear potential for multiple followup work”.

First, I find that the setup and coordinate system should be better described. • How exactly does the setup in Fig. 1 avoid the problem of single-particle tunneling from nodal gapless states in the parent superconductor?

— The referee’s question was addressed in the first paragraph of Sec 2 of our original manuscript: “this can be achieved by taking advantage of the fact that the single- particle tunneling and superconducting proximity effect have distinct spatial profiles: the former effect is peaked at the nodal direction and vanishes along the x and y directions, while for the latter it is the opposite. Thus, single-particle tunneling can be effectively suppressed by geometrically separating the diagonal portion QSH edge with the d-wave superconductor. We depict such a setup in Fig. 1.” To further clarify our idea, we have changed the last sentence to “we depict such a setup in Fig. 1, in which the corner region of the $d$-wave SC is rounded and spatially separated from the QSH layer.”

We further note that other than the d-wave setup we also mentioned other platforms recently proposed using fully gapped s-wave superconductors that is free from this issue.

• What is the coordinate system used? Is x a coordinate that parametrizes the position along the edge in Fig.1 (including the corner), or is the (x,y) coordinate system fixed with respect to the sample? It is not always clear to follow the conventions adopted throughout the text, especially when it comes to the field labeling.

— We thank the referee for pointing out this inconsistency in our notation. The referee is correct that the x appearing in the edge effective theory is a coordinate that parametrizes the edge including the corner. The (x,y) coordinate, for the magnetic field, on the other hand, is fixed for the sample but not necessarily in the plane of the sample, as we mentioned in Sec 2.1. They are instead two directions perpendicular to the conserved spin component in the effective theory.

To avoid future confusion, we have replaced the coordinates for the edge theory with $s$ and $s’$.

Second, the simplicity and robustness of the proposal are somewhat oversold and should be discussed more critically. • Most importantly, I do not understand the robustness of the "pie slice" protocol in Fig. 4. Even if all required symmetry conditions are satisfied, it seems to me that a small error in implementing the angle tau0 will result in a linear error in the final phase gate of Eq. 60. So how can the Berry phase be robust to small deformations of the arc? Practically speaking, what is the advantage with respect to a poor-man's rotation implemented via the timing of an interaction between MZMs?

— The referee is correct that to realize an arbitrary phase angle \tau_0, the input for \tau_0 needs to be precise; this is pointed out Sec 4.1.3. Understandably, to the best our knowledge, this is shared by every proposal that can realize an arbitrary phase gate (including the magic gate). For example in Ref. 21, three independent SC fields needs to be precisely manipulated along a complicated path to cancel out errors caused by non-universal coupling parameters.

However, one key merit of this setup, is that it is free from random errors caused by coupling to the environment due to the robustness of MZMs, and from unitary errors caused by non-universal coupling constants as long as the symmetries are preserved.

In light of the referee’s comments, we have emphasized in the introduction that unlike exchange gates that are completely protected by symmetries, for a phase gate the input for the phase angle needs to be precise.

• The effective time-reversal symmetry T_tilde seems to be crucial for protected operations. What are examples of perturbations that break this symmetry? Can they be expected in "real" devices?

— The effective time-reversal operator consists of the physical time-reversal operator with a U(1) spin rotation.

In terms of real devices, we note that in the 2d topological insulator WTe2, such a spin axis indeed exists and is recently determined in https://arxiv.org/abs/2010.09986. We have added this reference and a discussion on it in the revised version.

For this to hold, at least approximately in the low-energy effective theory, the spin-orbit coupling causing a spin texture of the edge states needs to be weak compared to pairing gap.

Besides the emergent time-reversal symmetry, the system is also required to have an effective particle-hole symmetry (for exchanging MZMs from different corners). While particle-hole symmetry is never exact for any real materials, in the low-energy edge theory it can be emergent via gating the sample to tune the chemical potential.

• The entire scheme seems rely on the absence of disorder and on the presence of several crystalline symmetry. Can it survive against, say, irregularities or non-uniformity in the sample shape? For instance, it seems to me that a certain relative orientation between the QSH insulator and the d-wave superconductor must be required to avoid the interaction between the nodal gapless excitations, but this aspect, while mentioned, is not clearly illustrated and looks worryingly fragile.

— We emphasize that our scheme for quantum gates does not rely on crystalline symmetries. Indeed, as we mentioned in Sec 2, in order to view the system with corner Majorana modes with nontrivial (higher-order) bulk topology, crystalline symmetries such as mirror reflection are needed. However, for our purposes, the model can be completely described by an edge theory with on-site symmetries only.

Regarding the suppression of nodal tunneling, we have tried to clarify this in a previous answer. Relevant to this question, the nodal direction does not need to be precisely aligned with the corner point for the proposed quantum gates, although the alignment enhances the efficiency of suppressing the tunneling.

Finally, the clarity of the presentation would benefit from one thorough proof-reading. Overall, the style and length of the work are appropriate and pleasant, but the notation is inconsistent or hard to trace at times, an [the rest of the sentence is missing — the authors]

—We thank the referee for praising the writing style and pointing out the need for further proof-reading. Other than improving the notation as requested, we have gone through the text again and fixed a number of typos.

Referee 2

In their work, the authors propose a way to perform arbitrary quantum gates by manipulating Majorana corner states in a second-order topological superconductor. The starting point is a time-reversal invariant second-order topological superconductor as proposed in Refs. [61,62] of the manuscript. In such a system, each corner of a rectangular sample hosts a Kramers pair of Majorana corner states. When an additional in-plane field is applied, time-reversal symmetry is broken and the Kramers partners can split in space. Importantly, however, the presence of an additional emergent time-reversal symmetry still prevents the two Majoranas at a given corner from hybridizing. The authors show that changing the magnetic field along certain closed paths in parameter space gives rise to non-Abelian Berry phases that realize non-trivial rotations in the degenerate ground state manifold accompanying the Majorana corner states. It is proposed to use these rotations to perform quantum gates on qubits encoded within the degenerate ground state manifold. The manuscript addresses a timely topic, is well-written, and appears to be technically sound.

— We thank the referee for the very positive evaluation on our work.

There are, however, several points that I hope the authors can comment on further. In particular, I have some concerns regarding the experimental feasibility and stability of the proposed manipulations. Furthermore, there are also some technical points that would benefit from an extended discussion. Explicitly, the points I would like to raise are the following: 1. While the authors advertise their proposal as experimentally feasible and advantageous compared to several other computational schemes, there are a number of issues that might arise in an experimental realization of the proposed setup: 1.1. The authors give a detailed discussion of the symmetries required to protect the Berry phases under consideration. However, these symmetry constraints appear to be very stringent (see, e.g., the conditions (1)-(3) listed on pages 18 and 19 of the manuscript) and I am wondering how well they will be fulfilled in any experimental implementation of the proposed setup. In particular, it seems that an out-of-plane component of the Zeeman field would immediately break the emergent time-reversal symmetry required to protect the proposed phase gates. Given the extensive (and experimentally non-trivial) manipulations of the in-plane Zeeman fields required to achieve the proposed quantum gates, it seems extremely challenging to completely avoid the accidental presence of a small out-of-plane component of the Zeeman field. This is even more so, since the Zeeman fields have to be local and are assumed to be non-zero only at a given corner where the manipulation is supposed to take place. But how is this possible with only in-plane B-fields which also need to satisfy div B=0?

— Indeed, our setup requires a U(1) spin rotation symmetry in the absence of Zeeman field, which is seemingly a stringent requirement even as an approximate symmetry. However, we note that in a recent experimental work arXiv:2010.09986, indeed such a symmetry was shown to exist for the edge states. We have added a discussion around this fact in Sec. 2, which we believe is helpful for the discussion of symmetry requirements. On the other hand, the referee is correct that an out-of-plane Zeeman coupling would destroy the Majorana zero modes, which may be challenging for experimental realization. However, we believe this important requirement is made clear by the symmetry requirement of our proposed platform.

The referee also raised the issue of the consistency of local Zeeman coupling with the Maxwell equation for the magnetic field. More precisely, the (Bx, By) coupling describes the coupling to the edge states, rather than a magnetic field filling the lab. While div B =0 indeed may dictate some out-of-plane magnetic fields in regions far away from the 2d system, their effects from below and above the 2d system cancel. At the symmetry level, as long as the applied magnetic field is perpendicular to the spin axis of the edge states, the symmetry requirement is satisfied.

1.2. The authors assume perfect experimental control over the applied Zeeman fields. In particular, it is assumed that the path along which the Zeeman field evolves in the (B_x,B_y) plane is always a closed one. However, what if---by accident---one does not end up at the same point in parameter space from which one has started, i.e., the path in parameter space is not closed? If the experimental control of the Zeeman field is not perfect, such errors could easily go unnoticed. There could also, for example, be a danger of ‘overshooting’ the magnetic field and performing a 2\pi\pm\epsilon rotation instead of an exact 2\pi rotation (in case of a full braid). How would this affect the results presented in the manuscript, and are there ways to detect, remedy, or avoid such errors?

— We thank the referee for this interesting question. From a theoretical perspective, a Berry phase is not well-defined if the starting and end point for a quantum state are different. In practice, as long as the symmetry requirements are satisfied, just like actually braiding and exchanging anyons, such imperfections only cause errors at the measurement stage of a quantum computation, where precise control is indeed necessary.

On the other hand, for the phase gate with an arbitrary \tau_0, indeed the execution of field rotation as an input for \tau_0, needs to be precise. Understandably this is shared feature of all proposals for arbitrary phase gates. We discussed this in more details in our response to Referee 1.

1.3. One should mention that a d-wave superconductor by itself already presents a highly non-trivial state of matter. It would be interesting to see if similar results could also be obtained from a time-reversal invariant topological insulator engineered from conventional ingredients only (see, e.g., Phys. Rev. B 102 (19), 195401).

— We thank the referee for pointing out the interesting reference, which we have included in the revised manuscript. Since our theoretical results come from the edge effective theory, the d-wave SC is not a necessary component and we see no immediate reason why it cannot be applied to other systems with proper emergent symmetries. In the manuscript we have also noted other proposals realizing corner Majorana pairs using Fe-based s-wave superconductors.

However, we would like to argue that the d-wave SC component in our proposed platform presents an advantage rather than a drawback. It is indeed highly nontrivial due to its perplexing phase diagram, but the d-wave pairing symmetry of the pairing phase is well established. It is also readily available experimentally, offering a high superconducting temperature and a large superconducting gap.

  1. When the model is studied in bosonic language, the considered situation is slightly different from the initial situation studied in the fermionic case, see Fig. 2. In the latter case, the length l depends on the strength of the Zeeman field and goes to zero as the Zeeman field goes to zero. In the model that is studied in the bosonic language, however, the length l is fixed and stays finite at all times. This is crucial for the expression for the level spacing given in Eq. (46) and as such also for stability of the proposed ‘half-moon’ and ‘slice-of-pie’ contours where the Zeeman field goes through zero at some point. The authors should comment on the differences between the two models studied in the fermionic and the bosonic language. Do these differences affect the results that are obtained and if so, in what way? Which situation is closer to the one actually encountered in an experimental realization of the model?

— This is a great question. The seeming disparity between the bosonic and fermionic languages is caused by our crude treatment of the bosonized edge theory. To be specific, instead of using a spatial depending mass term for the pairing gap and the Zeeman gap, we treated one of them as infinite beyond a small finite region, which sets a boundary condition. However, this treatment suits our purposes well, since we are only interested in the existence of zero modes of the theory. In particular, the length $\ell$ enters our result as an inverse gap scale for the zero modes, and the $\ell\to 0$ limit in our results are fully well-defined, even though the excited states are not captured.

In application to an actual experimental setting, one can obtain $\ell$ by direct measurement and use as input to the bosonic theory. For weakly interacting systems, the fermionic treatment should suffice.

  1. In the case of the ‘half-moon’ and ‘slice-of-pie’ contours, the Zeeman field goes to zero at some point of the process. In this case, the gap separating the ground state manifold from the excited states is given by the finite-size level spacing, see Eq. (46). I would expect that this gap can become quite small, which would then again put strong constraints on the adiabaticity of the process. Can the authors give an estimate on the energy scales involved in a realistic implementation of their scheme, and specify what this means for the adiabaticity condition of the process?

— Indeed the referee is correct that the gap is given by finite-size level spacing, which is inversely proportional to the size of the Zeeman field region and proportional to the Fermi velocity of the edge states. In lieu of real material parameters, a very crude estimate of this size is given by the penetration depth of the d-wave superconductor, e.g., a cuprate system. Thus, in order to maintain adiabaticity, the typical time scale spent in this region of the contour needs to be much longer the size of the Zeeman field region divided by the Fermi velocity.

  1. The intuition behind the ‘braiding’ process via a 2pi rotation of the in-plane Zeeman field could be made clearer. As far as I understand, there is no actual spatial exchange of the two Majoranas. Rather, the process seems to be closely connected to the one discussed in arXiv:1803.02173 (Ref. [58] in the manuscript). What is the intuition behind calling such a process an actual ‘braiding’ process? It would greatly benefit the accessibility of the manuscript if the authors could add some additional explanations here.

— We thank the referee for the great question. Indeed, it may seem counterintuitive that the effect of this process is identical to a physical braiding of Ising anyons (MZMs). Since the system under consideration is not a uniform topologically ordered state with anyonic excitations, we do not have an intuitive way to compare it to the actual braiding process. The best intuitive explanation is that the 2\pi rotation can be split to two \pi rotations (without the straight-line portions, which cancel out between the two), each of which does involve physical exchange of positions.

  1. It should be emphasized more clearly that some of the proposed manipulations (in particular, the arbitrary phase gates via ‘slice-of-pie’ contours) do not correspond to quasiparticle braiding processes. While the introduction correctly states that it is fundamentally impossible to realize non-Clifford gates by braiding Majoranas, this distinction is somewhat blurred in the main part of the manuscript. This is also due to the fact that even the single exchange (‘half-moon’ contour) does not correspond to a standard exchange of Majorana quasiparticles as one would typically have in mind when talking about topological quantum computation by non-abelian anyons. As the magnetic field goes to zero during the exchange process, the two Majoranas strongly overlap and the inherent topological protection accompanying a set of well-separated Majoranas is lost. What one is left with is merely a symmetry-protected non-Abelian Berry phase that could emerge also in any other two-level system (not based on anyonic quasiparticles) in the presence of suitable symmetries. Overall, the processes discussed in the manuscript at hand seem to be closely related to holonomic quantum computation. The authors should comment on these connections more transparently.

— Indeed, what we are proposing here is partly equivalent with, but not the same as the braiding anyonic quasiparticles. The distinction is manifested by the additional symmetry requirements. In response to the referee’s comments, we have added a clarifying sentence in the beginning of Sec 4.

  1. The results on the parafermionic case come with some unresolved problems such as unwanted dynamical phases. While this is correctly pointed out in the main text, I recommend that the corresponding part of the abstract should be phrased more carefully. In particular, the sentence “Our analysis naturally includes interaction effects and can be generalized to cases with fractional bulk excitations” tends to oversell the results on the parafermionic case.

— We thank the referee for pointing this out, and we have deleted the sentence on the parafermionic case in the abstract to avoid misleading the readers.

  1. Finally, I have a few more detailed remarks regarding specific parts of the manuscript: 7.1. Fig. 1: I do not understand the meaning of the red oval in the drawing. Does it represent a path in parameter space (=direction of magnetic field)?

— The inside region of the red oval denotes the region of the Zeeman field. Apologies for our limited artistic skills, but we have added a sentence clarifying this.

7.2. Page 6, in between Eqs. (2) and (3): I am not sure I understand what is meant by the remark ‘We choose x to be along the diagonal direction.’ From the equations, I would assume x to be an edge coordinate running along the edge of a finite sample. It would be helpful to indicate the coordinate axes in Fig. 1, for example.

— We meant “we choose the origin at the corner”. Following the suggestion of the other referee, we have changed the edge coordinate to $s$, and reserved $x,y$ for the magnetic field components. We hope this eliminates all potential confusions.

7.3. In Eq. (9): x_0 is not defined. Should it be x_-?

— Yes, fixed.

7.4. On page 7, after Eq. (9), it reads: “[…] there exist a pair of MZMs separated by a length l, the length of the region where |\Delta|=B.” Should it be |\Delta|<B?

— Yes, fixed.

7.5. Eq. (15): Is the complex conjugation on the RHS correct? It is already included in the definition of P.

— The referee is correct. There should be no complex conjugation. This has been fixed.

7.6. Page 8, after Eq. (16): I believe that \bar{C} is not defined.

— It should be just C. This has been fixed.

7.7. Eq. (18b): It should be x’ instead of x in the second argument on the LHS.

— Thanks, this has been fixed.

7.8. Eqs. (55), (56), and (57): I do not understand the difference between the \psi in the bra and \Psi in the ket. Is it a different state?

— It is a typo. They are the same state.

7.9. Page 18, condition (2): I believe there is a typo – the same condition Ba>>(a/l)^{2-K} is given twice. The same question also applies to page 19, again condition (2).

— Thanks for pointing this out. One of the $\gg$’s should have been $\ll$.

7.10. Eq. (68): The use of \tau_0 is somewhat unfortunate here, since the same symbol is also used for the zeroth Pauli matrix in particle-hole space [see, e.g., Eq. (10)].

— Thanks for pointing this out; we have changed $\tau_0$ in this context to $\theta_0$.

7.11. Eq. (70): I do not understand the meaning of (and the difference between) k_a, k_b, k_x, and k_y.

— This inconsistency came from an earlier version. We have replaced k_a with k_x and k_b with k_y.

7.12. Page 23, paragraph after Eq. (71): H_B is not defined. Should it be H_Z?

— Yes, this has been fixed.

---

## Round 2 · List of Changes

1. Clarified the setup, including the meaning of the red oval, in Fig.1
  2. Changed the coordinates in the edge theory to $s$.
  3. Clarified the requirement on the input for the phase gate.
  4. Added a new reference on the U(1) spin rotation (and hence \tilde T) for WTe_2.
  5. Added a few references on the realization of corner Majorana modes from s-wave superconductors.
  6. Clarified the relation and distinction with true braiding processes using non-abelian anyons.
  7. Deleted a misleading sentence in the abstract regarding parafermions.
  8. Fixed a number of typos.

---

## Editorial Decision

published